# HDX-MS reveals dysregulated checkpoints that compromise discrimination against self RNA during RIG-I mediated autoimmunity

Jie Zheng[1], Chen Wang[2], Mi Ra Chang[1], Swapnil C. Devarkar[3], Brandon Schweibenz[3], Gogce C. Crynen[4], Ruben D. Garcia-Ordonez[1], Bruce D. Pascal[1,5], Scott J. Novick[1], Smita S. Patel[3], Joseph Marcotrigiano[2] & Patrick R. Griffin [1]

Retinoic acid inducible gene-I (RIG-I) ensures immune surveillance of viral RNAs bearing a 5'-triphosphate (5'ppp) moiety. Mutations in RIG-I (C268F and E373A) lead to impaired ATPase activity, thereby driving hyperactive signaling associated with autoimmune diseases. Here we report, using hydrogen/deuterium exchange, mechanistic models for dysregulated RIG-I proofreading that ultimately result in the improper recognition of cellular RNAs bearing 7-methylguanosine and $N_1$-2'-O-methylation (Cap1) on the 5' end. Cap1-RNA compromises its ability to stabilize RIG-I helicase and blunts caspase activation and recruitment domains (CARD) partial opening by threefold. RIG-I H830A mutation restores Cap1-helicase engagement as well as CARDs partial opening event to a level comparable to that of 5'ppp. However, E373A RIG-I locks the receptor in an ATP-bound state, resulting in enhanced Cap1-helicase engagement and a sequential CARDs stimulation. C268F mutation renders a more tethered ring architecture and results in constitutive CARDs signaling in an ATP-independent manner.

[1] Department of Molecular Medicine, The Scripps Research Institute, Jupiter, FL 33458, USA. [2] Structural Virology Section, Laboratory of Infectious Diseases, National Institute of Allergy and Infectious Diseases, National Institutes of Health, Bethesda, MD 20892, USA. [3] Robert Wood Johnson Medical School, Rutgers University, Piscataway, NJ 08854, USA. [4] The Center for Computational Biology, The Scripps Research Institute, Jupiter, FL 33458, USA. [5] Omics Informatics LLC, Honolulu HI 96813, USA. These authors contributed equally: Jie Zheng, Chen Wang. Correspondence and requests for materials should be addressed to J.Z. (email: jzheng@scripps.edu) or (email: jzheng@simm.ac.cn) or to J.M. (email: joseph.marcotrigiano@nih.gov) or to P.R.G. (email: pgriffin@scripps.edu)

The innate immune system provides the first barrier against invasion of RNA viruses, which are sensed by cytosolic retinoic acid-inducible gene-I (RIG-I)-like receptors (RLRs), including RIG-I and melanoma differentiation-associated protein 5 (MDA5)[1–5]. However, a hyperactive and imbalanced innate immune system could result in undesirable autoimmune disorders regardless of viral attack[6]. Singleton–Merten syndrome (SMS) is a rare multisystem autoimmune disorder with clinic symptoms including aortic and valvular calcification, dental anomalies, osteopenia, or osteoporosis[7–11]. Previous studies have reported clinical phenotypes of classic and atypical SMS caused by missense mutations discovered in *IFIH1* (encodes MDA5)[8,12–14] and *DDX58* (encodes RIG-I)[7], indicating that elevated inflammation induced by RLR gain-of-function mutants contributes to the pathogenesis of SMS. In contrast, wild-type (WT) RIG-I and MDA5 are pattern recognition receptors (PRRs) of an innate immune system that ensure immune surveillance of RNAs with pathogen-associated molecular patterns (PAMPs)[15,16]. It, therefore, suggests that RLR proofreading avoids self-recognition by self-RNAs, whereas dysregulated proofreading leads to recognition of self-RNAs that could result in autoimmune diseases.

RIG-I and MDA5 consist of two amino-terminal tandem caspase activation and recruitment domains (CARD1-2) that are important for signal transduction; a central DExH/C HELicase domain (HEL); and a carboxyl-terminal domain (CTD)[17–19] (Fig. 1a). As a member of duplex RNA-activated ATPases, RLR shares a superfamily II (SF2) helicase core that contains two RecA-like domains (HEL1-2) and an insertion domain HEL2i[17] (Fig. 1a). RIG-I and MDA5 signal through their common adapter

protein mitochondrial antiviral signaling (MAVS), which possesses a CARD domain at its N terminus[20,21]. Upon activation, RIG-I or MDA5 CARDs interact with MAVS driving nucleation of MAVS filament formation[19,21–24]. Non-covalent K63-linked polyubiquitin plays a critical role in this activation process. Atomic resolution structures have been previously determined that provide the molecular basis for MAVS filament formation, including a poly-ubiquitin chain-linked RIG-I CARDs tetramer and a RIG-I CARDs-MAVS CARD complex[19,24]. The activated MAVS fiber initiates signaling cascades leading to upregulation of type I interferon (IFN)[5,25–27].

Comparing to MDA5, the molecular features of non-self and self-RNAs leading to recognition and discrimination by RIG-I are better characterized[15,16]. Duplex RNAs with 5′-triphosphate moieties (5′ppp) are the PAMP RNAs frequently found in viral RNA genomes or replication intermediates[28,29]. Unlike PAMP RNAs, host or self-messenger RNAs (mRNAs) of humans and other higher eukaryotes possess 7-methyl guanosine (m7G) cap linked to the γ phosphate of a triphosphate at the 5′ end (termed as Cap0)[30] (Fig. 1b). In addition, 2′-O-methyl modifications at first nucleotide ribose ($N_1$ position, termed as Cap1) or first and second nucleotides ($N_1$ and $N_2$ positions, termed as Cap2) are conserved molecular features associated with mRNAs in humans and other higher eukaryotes[1,31,32] (Fig. 1b). Cap1 or Cap2 RNA critically prevents immune recognition by RIG-I in cell-based IFN assays and therefore considered as a molecular marker for self-RNAs[15]. WT RIG-I discriminates against Cap1 by a mechanism involving specific recognition of the $N_1$-2′-OH in the RNA backbone by the H830 in the CTD[15]. PAMP RNA (5′ppp) binds RIG-I with a high binding affinity that facilitates the formation of

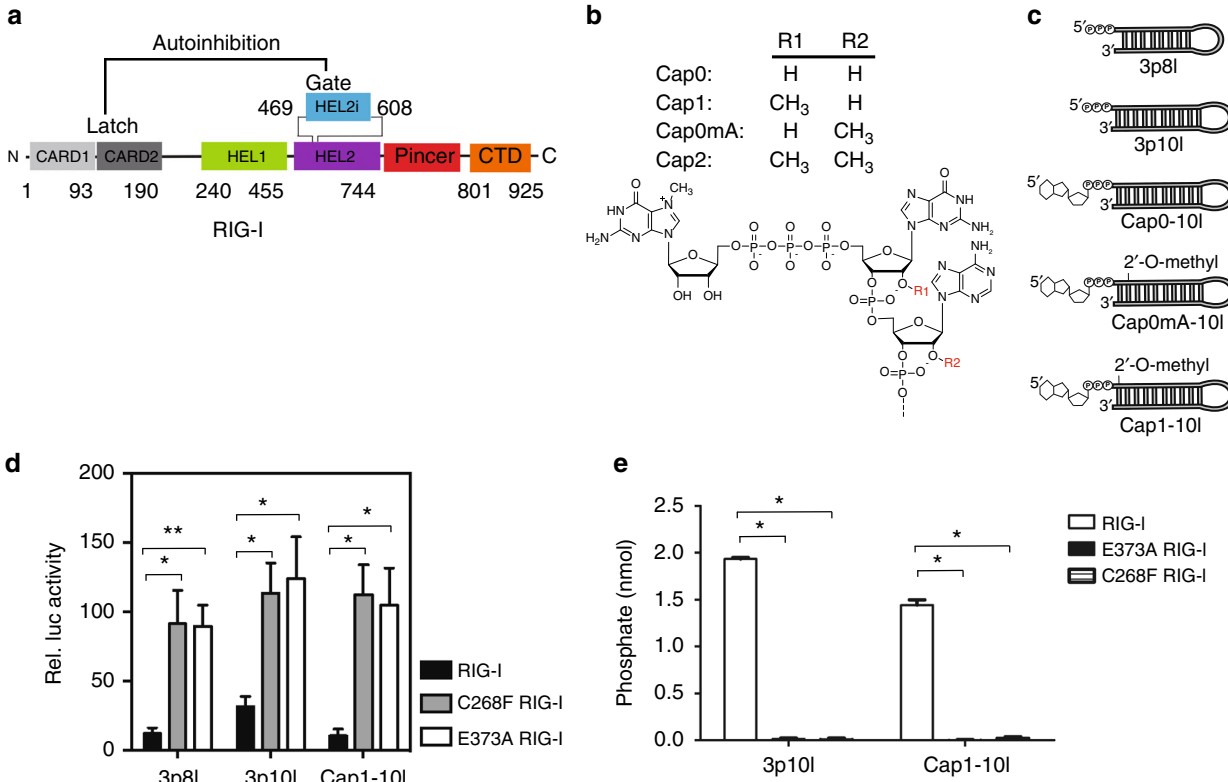

**Fig. 1** RNA chemical structures and IFN-β activity. **a** Domain arrangement of RIG-I (1–925). Autoinhibition is shown between CARD2 latch region and HEL2i gate motif. **b** The chemical structures of an m7G cap and 2′-O-methylation modification at $N_1$ or $N_2$ position are presented. Important features for Cap0, Cap1, Cap0mA, and Cap2 are highlighted. **c** Schematic representations of studied 8- and 10-mer hairpin RNAs that bear different modifications at the 5′ RNA terminus. 3p10l and Cap1-10l represent the molecular signature of viral and self-RNA, respectively. **d** The IFN-β luciferase signal is plotted as recorded in WT/SMS RIG-I dual reporter assays stimulated with indicated RNAs. Basal IFN-β activity of WT RIG-I in the absence of RNA transfection was subtracted. The significance of differences between groups was evaluated by unpaired Student's *t* test (*$p < 0.05$; **$p < 0.01$). **e** ATPase activities of Cap1-10l- and 3p10l-bound WT/SMS RIG-I. The significance of differences between groups was evaluated by unpaired Student's *t* test (*$p < 0.05$)

an active ATPase cleft within the SF2 ATPase core, enabling RIG-I to robustly bind and hydrolyze ATP[1,4].Yet, Cap1 RNA has a weak binding affinity and reduced ATP hydrolysis activity[1]. Although RIG-I may discriminate between 5′ppp RNA and Cap1 RNA by affinity, the contributions of m7G cap and $N_1$-2′-O-methylation on RIG-I CARDs activation remains to be understood.

Gain-of-function SMS RIG-I mutations—E373A and C268F—lead to increased type I IFN levels in the absence of virus invasion[7,33]. Several studies have reported the existence of self-RNAs that could be recognized by RIG-I in cellulo and in vivo. For instance, one study has performed RIG-I RNA immunoprecipitation from cell lysates of the murine splenic B-cell line and provided direct evidence that WT RIG-I recognizes several regions within nuclear factor-κB1 (NF-κB1) 3′-untranslated region mRNA[34]. Lassig et al.[35] have performed immunoprecipitation of RIG-I-RNA complex from virus-infected and non-infected HEK293T RIG-I knockout (KO) cells, which showed the interaction of RIG-I ATPase-deficient mutant E373Q with host ribosomal RNAs. Similarly, an increased amount of RNA is reported to be co-purified from C268F and E373A RIG-I from uninfected cells compared to that of WT RIG-I[35]. Schuberth-Wagner et al.[15] showed that cellular RNAs could activate H830A mutant RIG-I but not WT RIG-I. Another study using an in vivo murine model validated that small self-RNA fragments generated by RNase L can trigger IFNβ responses via the RIG-I signaling pathway[36]. Interestingly, E373 and C268 residues are within the ATPase cleft with the former residue residing on HEL1 motif II the latter sitting on HEL1 motif I, and mutations of these residues render RIG-I ATPase-deficient and signaling constitutively active[33]. In SMS variant-transfected human embryonic kidney 293T (HEK293T) cells, sufficient phosphorylation level of IFN regulatory factor-3 (IRF3) and IRF3 dimerization have been observed at the basal level. Expression of SMS RIG-Is have been shown to be associated with significantly enhanced NF-κB reporter gene activity and increased expression of interferon β gene (IFNB1) and interferon-stimulated gene 15 (ISG15) in both unstimulated and polyinosinic:polycytidylic acid-treated cells[7]. However, the molecular mechanisms by which SMS RIG-I mutants become activated by self-RNA, while WT receptor is unaffected remains unclear.

Hydrogen/deuterium exchange coupled with mass spectrometry (HDX-MS) has emerged as a sensitive and robust method to study solution-phase protein dynamics perturbed by different binding events, such as those by chemical ligands, nucleic acids, or co-regulator proteins[4,37]. Previously, HDX-MS provided direct evidence that apo RIG-I adopts gated conformation via CARD2-HEL2i engagement[4]. This intra-molecular interaction governs the threshold of CARDs allosteric release, which depends on the extent of signal transmission during RNA recognition and ATP binding events by RIG-I[4].

Here, we employ HDX-MS to provide new mechanistic insights into aberrant CARDs activation during dysregulated RIG-I proofreading of self-RNAs. E373A and C268F mutations differently affect RIG-I proofreading via an ATP-dependent and -independent manner, respectively. Our results could also explain other autoimmune diseases related to innate immunity.

## Results

**RNA chemical structures and IFN-β activities**. To specifically examine the impact of m7G cap and 2′-O-methylation on RNA surveillance by receptor, 10 base pair (bp) hairpin duplex RNAs (5′-GAAUAUAAUAGUGAUAUUAUUAUUC-3′) were synthesized with a 5′ppp (3p10l), a triphosphate moiety capped with m7G (Cap0-10l), Cap0 with 2′-O-methylation modified at the first nucleotide ribose guanine (Cap1-10l) or Cap0 with 2′-O-methylation modified at the second nucleotide ribose adenine (Cap0mA-10l) (Fig. 1b, c). In addition, 5′ppp RNA with stem

**Table 1 RNA binding and ATPase activity of WT RIG-I**

| RNA molecules | $K_{d,\ app}$ (nM) | $k_{atpase}$ (s$^{-1}$) |
|---|---|---|
| 3p8l[a] | 0.6 ± 0.3 | 1.2 ± 0.02 |
| 3p10l[b] | 1.8 ± 0.9 | 33 ± 0.9 |
| Cap0-10l[b] | 1.7 ± 0.5 | 25 ± 0.4 |
| Cap0mA-10l[a] | 2.4 ± 0.8 | 18.3 ± 0.3 |
| Cap1-10l[b] | 425 ± 50 | 15 ± 0.7 |

[a] The data are related to Supplementary Fig. 3
[b] The data are derived from reference [1]

region of 8bp (5′ppp-GGCGCGGCUUCGGCCGCGCC-3′, termed 3p8l) was used as an inactive RNA molecule incapable of eliciting RIG-I-mediated IFN responses in cellular assays (Fig. 1b, c)[38,39]. Comparable with 3p10l and Cap0-10l, Cap0mA-10l binds to WT RIG-I with sub-nanomolar affinities and activates RIG-I's ATPase activity (Table 1). In contrast, Cap1-10l binds to RIG-I with 200-fold lower affinity, suggesting that 2′-O-methylation at the second nucleotide is not critical for RIG-I-mediated RNA discrimination. In WT RIG-I-mediated IFN reporter assays, 3p10l stimulated a 3-fold higher IFN-β signal response compared to that observed with 3p8l and Cap1-10l RNAs (Fig. 1d). These observations suggest that Cap1-10l, as a self-RNA representative, behaves differently from PAMP-RNAs with respect to RNA discrimination by RIG-I and CARDs activation. Significantly higher IFN-β reporter signal response was observed in SMS RIG-I-mediated signaling pathways regardless of Cap1 or 5′ppp or even inactive RNA (Fig. 1d). In vitro studies further indicate that SMS RIG-Is are ATPase deficient (Fig. 1e). Therefore, SMS RIG-Is (C268F and E373A) possess constitutively active signaling, yet have compromised ATPase activity.

**Cap1 discrimination by HEL-CTD RNA recognition module**. Digestion optimization for HDX studies resulted in >90% sequence coverage for full-length WT RIG-I and gain-of-function RIG-I mutants H830A, E373A, and C268F (Supplementary Fig. 1a, b and c). In the absence of fragmentation data from electron capture dissociation, the residue level deuteration data were approximated by using HDX data from overlapping peptides and consolidating these data using a residue averaging approach as previously described[40] (Supplementary Fig. 1b). These data were mapped to the structure using HDX Workbench and PyMOL[41,42]. The presence of 5′ppp and 5′m7Gppp (Cap0) moieties at the ends of the 10 bp hairpins—3p10l and Cap0-10l—reduced HDX and increased stabilization of the hydrogen bond networks within the CTD capping loop (V843–856) (~12% reduction in HDX relative to apo RIG-I). Whereas the same region of the protein was unperturbed in the presence of Cap1-10l RNA (Supplementary Fig. 1a, b and c, column (iii), (iv), and (v) and Supplementary Fig. 2a). In contrast, in the presence of Cap0mA-10l (with a 2′-O-methylation modification at the $N_2$ position of the RNA), the protection to HDX (~13% change compared to no RNA) in the CTD capping loop was comparable to Cap0-10l and 3p10l RNAs (Supplementary Fig. 1a, b and c, column (vi) and Supplementary Fig. 2a). Although 3p8l appeared inactive in the RIG-I IFN-β assay[39], it showed increased stability in the CTD capping loop (~21% increased protection to exchange as compared to an apo receptor) (Supplementary Fig. 1a, b and c, column (vii) and Supplementary Fig. 2a). These results indicate that an m7G cap alone does not perturb the 5′ppp-CTD capping loop interaction, but the 2′-O-methylation at the $N_1$ position, and not the $N_2$ position, is critical in RNA discrimination by the RIG-I CTD capping loop. Various RNA-binding motifs, including HEL1 motif Ia, Ic, IIa, HEL2i (V522–539), and CTD RNA-binding motif V893-904, showed reduced HDX upon recognition of 3p10l, Cap0-10l, and Cap0mA-10l molecules, but this was not

observed with Cap1-10l (Supplementary Fig.1 a, b and c, column (iii), (iv) and (vi)). Only the HEL1 motif Ia and CTD RNA-binding surface (AA893–904) exhibited increased protection to HDX with Cap1-10l RNA, indicating limited RNA engagement with the HEL-CTD RNA recognition module (Supplementary Fig. 1a, b and c, column (v)). These data are consistent with the 200-fold decrease in binding affinity between RIG-I and Cap1 RNA (Table 1). These observations suggest that, unlike other 10 bp hairpins, Cap1 representing "self-RNAs" engage the RIG-I CTD capping loop and HEL-CTD RNA recognition module differently. Furthermore, comparison between 3p10l and 3p8l bound helicase-RD RIG-I X-ray crystal structures[38] and their HDX footprints reveal that HEL2i is engaged with 10 bp hairpin and not the 8 bp hairpin RNA, showing its ability to sense the length of RNA duplex[39] (Supplementary Fig. 1a, b and c, column (iii) and (vii)). It is also known that H830 in the CTD interacts with $N_1$-2′-OH and therefore discriminates against $N_1$-2′-O-methylation within Cap1 RNA[1,15]. To understand the mechanism of RNA surveillance by H830, we first examined HDX perturbations in the CTD capping loop upon H830A RIG-I binding to 3p10l and Cap1-10l RNA, respectively. Similar to the CTD capping loop of WT RIG-I, the H830A RIG-I CTD capping loop maintained sensitivity towards 5′ppp moiety (~19% increase protection to HDX), whereas the same region in the mutant RIG-I displayed apo-like conformational dynamics upon Cap1-10l recognition (Supplementary Fig. 1a, b and c, column (xiii) and (xiv) and Supplementary Fig. 2b). H830A RIG-I exhibited nearly identical HDX dynamics when bound to 3p10l and Cap1-10l, including the HDX footprints of the RNA-binding motifs, HEL1 motif Ia, Ic, IIa, HEL2i (V522–539) and the CTD RNA-binding motif V893–904. (Supplementary Fig. 1a, b and c, column (xiii) and (xiv)). These data are consistent with the previous observation that H830A substitution restores high RNA-binding affinity with Cap1 RNA in a similar extent to that of a 5′-ppp RNA[1].

**RNA surveillance is coupled with CARDs partial opening**. Disruption of CARD2-HEL2i engagement is the first step of RIG-I-gated activation[4,43] and overexpression of CARDs module alone is enough to induce IFN signaling[44]. Our previous HDX study demonstrated that binding of PAMP RNA to RIG-I induces a conformational switch from stable auto-inhibited to destabilized CARDs conformation and binding of PAMP RNA alone is insufficient to fully expose the CARDs[4]. The CARD2 latch region (Y101–114), which is spatially locked to the HEL2i gate motif in the apo-form, displays EX1 exchange behavior upon binding to PAMP RNA, suggesting that this region undergoes a partial opening event and structural transition that facilitates correlated exchange in the protein[4]. If a refolding event occurs sufficiently slow to allow complete deuterium exchange of backbone amide hydrogens within the unfolding region, EX1 kinetics is observed[45]. Under EX1 conditions, if an opening or unfolding event involves more than one slow exchanging amide hydrogen, then deuterium exchange occurs simultaneously at these amides. Therefore, a bimodal distribution occurs via a correlated exchange pattern, in which the lower mass envelope corresponds to molecules that have not yet exchanged (not yet unfolded) and the higher mass envelope corresponds to molecules that have undergone exchange (molecules that have unfolded)[4,46–48]. The region undergoing EX1 kinetics could reveal the rate at which the relative proportions of unfolded and folded conformers interconvert. In contrast, EX2 kinetics takes place if refolding rate is much faster than the intrinsic exchange rate of the amide hydrogens, resulting in one isotopic envelope throughout the labeling time of the experiment.

We analyzed EX1 exchange kinetics of CARD2 latch peptide to probe RIG-I CARDs displacement during RNA discrimination by RIG-I proofreading. The appearance of EX1 kinetics under native state conditions is intriguing and could reveal conformational

intermediate state as well as the transition between two conformers[4,37,46,48,49]. As such, characterization of HDX dynamics of CARD2 latch peptide is critical to examine RNA-mediated RIG-I agonism, the different extent of CARDs disassembly, and perhaps more importantly RIG-I activation. In apo RIG-I, CARD2-HEL2i intra-molecular interaction greatly restricted CARDs deuterium uptake (Fig. 2a, b and Supplementary Fig. 1 a, band c, column (i)). The EX1 kinetic regime of CARD2 latch peptide in apo RIG-I was difficult to detect as there was very little deuterium exchange in the auto-inhibited domain even at the 1 h HDX time point (Fig. 2a). In contrast, analysis of the CARDs (1–228) protein, where the CARDs domain cannot be auto-inhibited, resulted in significant deuterium incorporation and the same latch peptide showed the absence of EX1 kinetics in the recorded HDX time points (Fig. 2a, b and Supplementary Fig. 1a, b and c, column (ii)). As an inactive RNA ligand demonstrating basal IFN signaling in cells, binding of 3p8l RNA to RIG-I failed to perturb CARD2-HEL2i interaction in the recorded HDX time points and presented apo-like isotopic distributions in Y103–114 motif (Fig. 2a and Supplementary Fig. 1 a, b and c, column (vii)). Interestingly, CARD2-HEL2i interface was disrupted by the PAMP RNAs, and the CARD2 latch peptide displayed differential EX1-like behavior upon Cap1-RNA and PAMP RNA binding (Fig. 2a). This suggests that RNA binding to RIG-I allosterically triggers partial opening of the CARDs, and the extent of opening in solution is dependent on the efficacy of specific RNA (Fig. 2a). The observed EX1 kinetics result in an increase in the intensity of a second isotopic envelope at higher $m/z$ value over time. This could be indicative of the rate of unfolding or opening of the cooperatively exchanged amides within the Y103–114 motif.

To more precisely measure the differential EX1 kinetics associated with Cap1-RNA and PAMP RNAs, we determined the CARDs structural transition rate from the lower MS envelope to the higher MS envelope by fitting two Gaussian peaks to the bimodal isotope cluster and dividing the entire peak area by the area of higher MS envelope or the isotope cluster area of the unfolded state (U/(U + F))[47,48]. An exponential 3P with the prediction model: $a + b \times \exp(c.\text{time (min)})$ could further accommodate the ratio of unfolded state in each data point and calculate the half-life ($t_{1/2}$) of CARDs partial opening. As shown in Fig. 2c, d, PAMP-like RNAs—3p10l, Cap0-10l, and Cap0mA-10l—exhibited similar transition rates in CARD2 latch peptide ($t_{1/2}$ values = 13.0 ± 1.2, 13.9 ± 1.2, and 12.9 ± 1.2 min, respectively). Analysis of transition rate in the same region of Cap1-10l-bound RIG-I revealed a nearly 3-fold delay with a $t_{1/2}$ value of 42.5 ± 4.3 min (Fig. 2c, d). The impact of 3p8l on CARD2 latch peptide dynamics was further examined at longer deuterium exchange time points. Binding of 3p8l to RIG-I induced protection against HDX in motif Ic and CTD RNA-binding regions throughout the longer HDX time points (Supplementary Fig. 2c). The CARD2 latch region showed enhanced EX1 behavior associated with 3p8l-bound RIG-I at the 3, 5, and 7 h time points (Supplementary Fig. 2d). The calculated $t_{1/2}$ value was much longer (4.97 ± 0.15 h) compared with those for other RNA ligands (Supplementary Fig. 2e and f). Although these observed differential EX1 kinetics and derived $t_{1/2}$ values cannot directly reveal RIG-I CARDs activation, they could be correlated with the different extent of CARDs partial opening events in solution and consistent with RNA-mediated RIG-I agonism. In contrast, RIG-I H830A exhibited comparable transition rates in the CARD2 Y103–114 region upon Cap1 and triphosphate RNA binding with $t_{1/2}$ values of 17.5 ± 1.0 and 17.6 ± 0.9 min, respectively (Fig. 2e, f). Additionally, comparable IFN-β reporter signal was observed in H830A RIG-I-mediated signaling pathway upon 5′ppp and Cap1 RNA stimulation (Fig. 2g). These data further demonstrate that

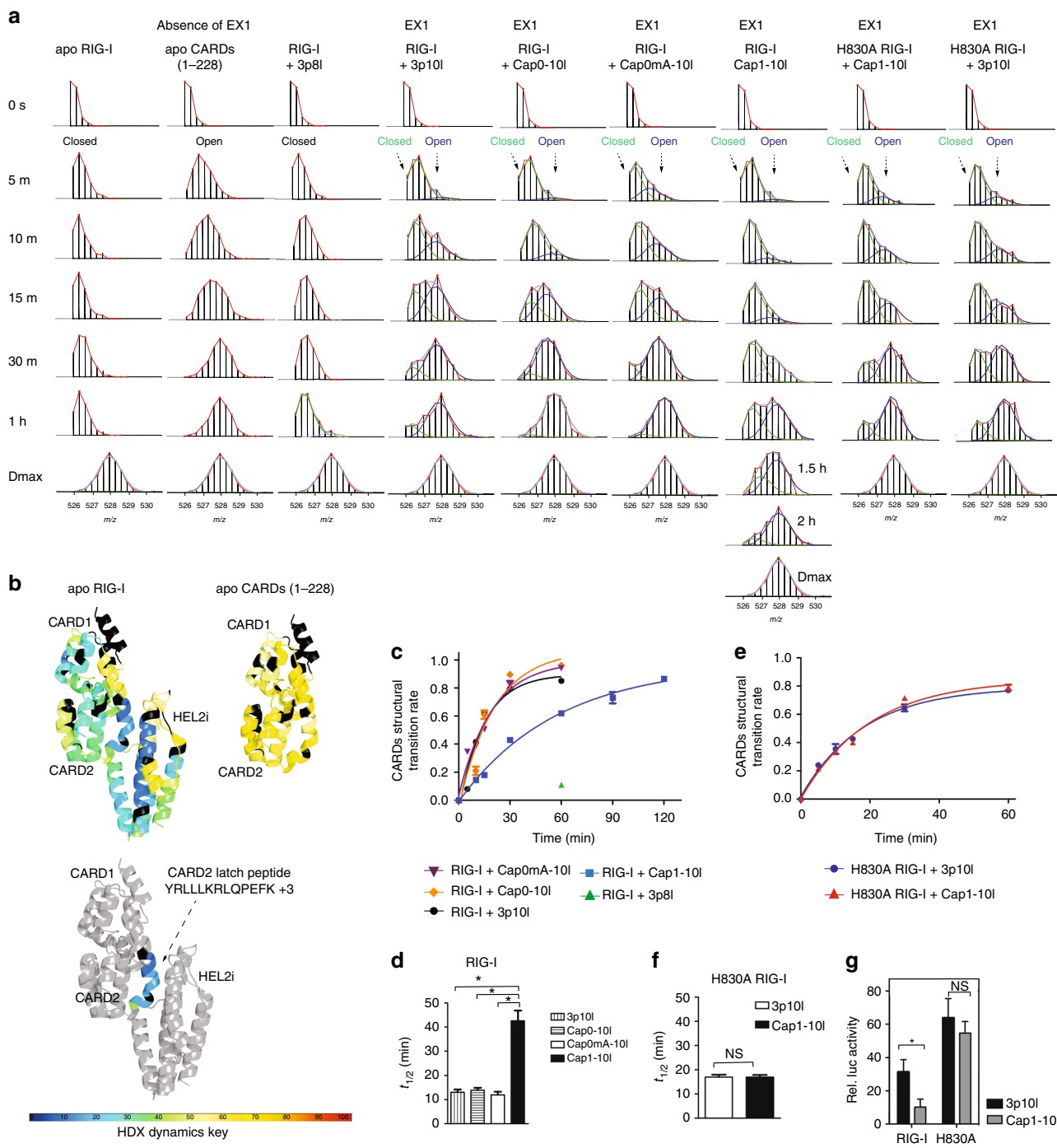

**Fig. 2** Partial opening of CARDs upon RNA surveillance by WT and H830A RIG-I. **a** MS spectra of RIG-I CARD2 latch peptide Y103–114 derived from various complexes at the indicated on-exchange time points. **b** HDX single amino acid consolidation view of *apo* RIG-I (Supplementary Fig. 1a, b and c, column (i) and (ii)) are, respectively, mapped to the CARDs-HEL2i structure model (upper left panel) and single CARDs structure (upper right panel), representing auto-repressed and solvent-exposed CARDs conformational dynamics. The location of CARD2 latch peptide is highlighted in the structure model (below). Percentages of deuterium uptake are color coded according to HDX dynamics key. Black, regions that have no sequence coverage and include proline residue that has no amide hydrogen exchange activity. **c** and **e** The fraction of WT or H830A RIG-I CARDs molecules in the higher MS population (open conformation) to the total population is plotted against the on-exchange time points. **d**, **f** Half-life ($t_{1/2}$) of respective partial opening event is determined by fitting an exponential 3P with the prediction model: $a + b \times \exp(c.\mathrm{time(min)})$, where $a$ is the asymptote, $b$ is the scale, and $c$ is the growth rate, is used to fit a curve to %D (response) and time (regressor). Inverse prediction is used to solve for the half-life ($t_{1/2}$) for each conformational state. (*Means that Cap1-10l-treated group predicted $t_{1/2}$ (42.53 min) was higher than the upper limit (24.67, $\alpha = 0.05$), whereas 3p10l ATP, Cap0-10l, and Cap0mA-10l-treated groups did not exceed the lower or upper limits in this comparison; NS means statistically nonsignificant between compared group. $t_{1/2}$ calculated for 3p10l-bound H830A RIG-I and Cap1-10l-bound H830A RIG-I did not exceed the lower or upper limits ($\alpha = 0.05$) in this comparison.) **g** The IFN-β luciferase signal is plotted as recorded in WT/H830A RIG-I dual reporter assays stimulated with indicated RNAs. The significance of differences between groups was evaluated by Student's *t* test (*$p < 0.05$; **$p < 0.01$)

H830 contributes to a crucial checkpoint during RNA discrimination by RIG-I.

**Cap1 and 5′ppp are distinguished by E373A RIG-I.** To probe the mechanism of dysregulation of RIG-I proofreading, we produced SMS RIG-I E373A mutant to examine the role of RIG-I E373 residue within HEL1 motif II (Walker B motif)[7]. Apo E373A RIG-I adopted similar auto-inhibited conformation as that of WT (Supplementary Fig. 1a, b and c, column (xv)). Similar to that of WT, binding of 3p10l to RIG-I E373A resulted in robust protection to HDX in HEL1 motif Ia, Ic, HEL2i (V522–539), CTD capping loop and CTD RNA-binding region (V893–904) (Supplementary Fig. 1a, b and c, column (xvi)). In contrast, only limited RNA-binding surfaces—HEL1 motif Ia and CTD V893-904 region—were protected to HDX upon E373A mutant binding to Cap1 RNA (Supplementary Fig. 1a, b and c, column (xvii)). These observations suggest that RIG-I E373A can distinguish triphosphate and Cap1 RNAs via the H830 mechanism. To examine the CARDs partial opening during the RNA sensing step, EX1 analysis of 3p10l-bound and Cap1-10l-bound RIG-I E373A complexes revealed different degrees of CARDs dissociation (Fig. 3a, b). There was more than a 3-fold increase in the transition rate from closed to open CARDs conformation in the former complex ($t_{1/2} = 20.5 \pm 0.9$ min) compared to the latter ($t_{1/2} = 72.8 \pm 17.8$ min) (Fig. 3a, b, d). Only subtle perturbation in HDX was observed in the CARDs domain of RIG-I E373A when bound to 3p8l suggesting the domain remains locked to the HEL2i gate motif in the presence of this RNA (Supplementary Fig. 1a, b and c, column (xviii)).

**E373A affects the ATP-dependent proofreading of RIG-I.** Apart from RNA recognition, activation of RIG-I is also coupled with ATP binding and hydrolysis, a critical checkpoint contributing to RIG-I proofreading[4,33,50,51]. E373 is conserved among SF2 helicase family members and coordinates $Mg^{2+}$ ion with ATP to form a functional $Mg^{2+}$-ATP complex[18]. To test whether the substitution of alanine with glutamate potentially impairs RIG-I ATP-binding and hydrolysis activity, we compared WT RIG-I bound to either 3p10l or Cap1-10l with E373A RIG-I bound to the same RNAs in the presence and absence of ATP. In the absence of ATP, both WT and E373A showed similar solvent protection patterns. In the presence of ATP, only E373A showed increased protection to HDX in HEL1 motif Q, motif Ic, HEL2i V542–549, HEL2 motif IVa, motif VI, and CTD RNA-binding region (I897–904) both in Cap1-10l and 3p10l bound complexes (Supplementary Fig. 1a, b and c, column (xix) and (xxi)). Interestingly, the CTD capping loop of E373A RIG-I showed significant protection to HDX (22 and 17% reduction) upon ATP addition in both 3p10l and Cap1-10l complexes, respectively, a phenomenon also not observed with WT RIG-I (Fig. 3f, g and Supplementary Fig. 1a, b and c, column (viii), (x), (xix) and (xxi)). These observations are intriguing, because ATP binding appears to allosterically transmit signals to E373A RIG-I CTD capping loop for enhanced interaction with RNA 5′ terminus regardless of different capped structures, which could increase the recruitment and local concentration of self-RNAs presented to the SF2 helicase core (Fig. 3h). By contrast, the CTD capping loop of WT RIG-I remained insensitive under ATP hydrolysis conditions (Fig. 3i).

Interestingly, ATP binding drove deuterium uptake in CARDs and HEL2i regions of E373A complex with Cap1-10l, suggesting that CARDs is further liberated into the solvent (Fig. 3a, b, d and Supplementary Fig. 1a, b and c, column (xxi)). However, this was not observed in the CARD2-HEL2i region of Cap1-bound WT RIG-I under ATP hydrolysis conditions (Fig. 3a, c, e and Supplementary Fig. 1a, b and c, column (x)). Further analysis of the bimodal distribution of CARD2 Y103–114 Cap1-RIG-I E373A complex with ATP revealed an accelerated transition rate

(an ~4.5-fold increase) of CARDs partial opening ($t_{1/2} = 15.9 \pm 0.8$ min) in comparison to without ATP ($t_{1/2} = 72.8 \pm 17.8$ min) (Fig. 3a, b, d). These observations suggest that ATP-binding energy spurred E373A receptor bound to Cap1-10l to displace CARDs ($t_{1/2} = 15.9 \pm 0.8$ min) at a rate similar to that observed with 3p10l with ATP ($t_{1/2} = 12.1 \pm 0.6$ min) (Fig. 3a, b, d). This sequential CARDs destabilization was also observed with WT/ E373A RIG-I-RNA complexes in the presence of the non-hydrolysable ATP analog ADP.AlFx (Supplementary Fig. 1a, b and c, column (ix), (xi), (xx) and (xxii)). Whereas the EX1 kinetics in Cap1-10l bound WT RIG-I and Cap1-10l-WT RIG-I-ATP complex were similar (Fig. 3a, c, e).

The non-hydrolysable ATP analog, ADP.AlFx, resembles the hydrolytic transition state due to the planar geometry of the γ-phosphate mimetic (AlFx). The presence of ADP.AlFx increased RNA engagement in the 3p10l-bound WT RIG-I complex within the RNA recognition module, resulting in significant protection to HDX in HEL1 motif I, Ia, Ic, IIa, III, HEL2i (V522–539), HEL2 (V600–613), motif IV, V, VI, Pincer region (F739–755), and CTD Y905–922 region (Supplementary Fig. 1a, b and c, column (ix)). This observation is consistent with crystallographic data of the RIG-I HEL-CTD showing that with RNA alone, HEL2 is mostly disordered with a relatively open conformation, and the presence of the non-hydrolysable ATP analog enables the receptor to adopt a more compressed conformation via robust engagement with HEL1, HEL2, Pincer, and CTD[1,17,18]. Such ADP.AlFx-dependent structural rearrangement within the HEL-CTD module transforms the WT RIG-I into an ATP-competent state and allosterically results in accelerated CARDs opening into the solvent, as revealed by increased deuterium incorporation in CARDs of ADP.AlFx-3p10l-WT RIG-I complex in comparison to 3p10l-WT RIG-I complex (Supplementary Fig. 1a, b and c, column (ix)). Larger buried areas with a higher degree of HDX protection were observed within Cap1 associated WT RIG-I complex in the presence and absence of ADP.AlFx (Supplementary Fig. 1a, b and c, column (xi)); correspondingly, more enhanced hydrogen bonding was observed in the CARDs region (Supplementary Fig. 1a, b and c, column (xi)). Similar to that of WT receptor, ADP.AlFx binding also greatly compressed CTD-HEL and dislodged CARDs to a similar extent in both ADP.AlFx-3p10l-E373A RIG-I and ADP.AlFx-Cap1-10l-E373A RIG-I complexes (Supplementary Fig. 1a, b and c, column (xx) and (xxii)). Taken together, these data suggest that the E373A substitution impairs ATP hydrolysis and enables the mutant receptor to adopt an ATP-bound state in the presence of ATP, which drives enhanced CARDs dissociation upon binding to self-RNAs such as the Cap1 RNA. In contrast, the ATP hydrolysis function of WT RIG-I serves as a checkpoint for effective RNA discrimination, preventing elevated CARDs opening and permitting the receptor to distinguish Cap1 from triphosphate.

**C268F RIG-I becomes signaling competent independent of ATP.** Although C268 resides in HEL1 motif I and is not implicated in ATP-binding activity[18,33], RIG-I C268F is known to be constitutively active with high IFN production via an unknown mechanism[7,33]. HDX analysis of apo C268F RIG-I confirmed its auto-inhibited conformation in solution (Supplementary Fig. 1a, b and c, column (xxiii)). Surprisingly, large buried surface areas within HEL-CTD module—HEL1 Ia, Ic, IIa, III, HEL2i V452–459, V522–539, HEL2 motif IV, V, VI, and CTD RNA-binding regions F838–904—exhibited identical HDX footprints in both Cap1-10l- and 3p10l-bound RIG-I C268F complexes (Fig. 4a, b and Supplementary Fig. 1a, b and c, column (xxiv) and (xxv)). This HDX signature is highly reminiscent of that observed for the RNA-RIG-I complex perturbed by ADP.AlFx, in which HEL-CTD formed a compact conformation and remained

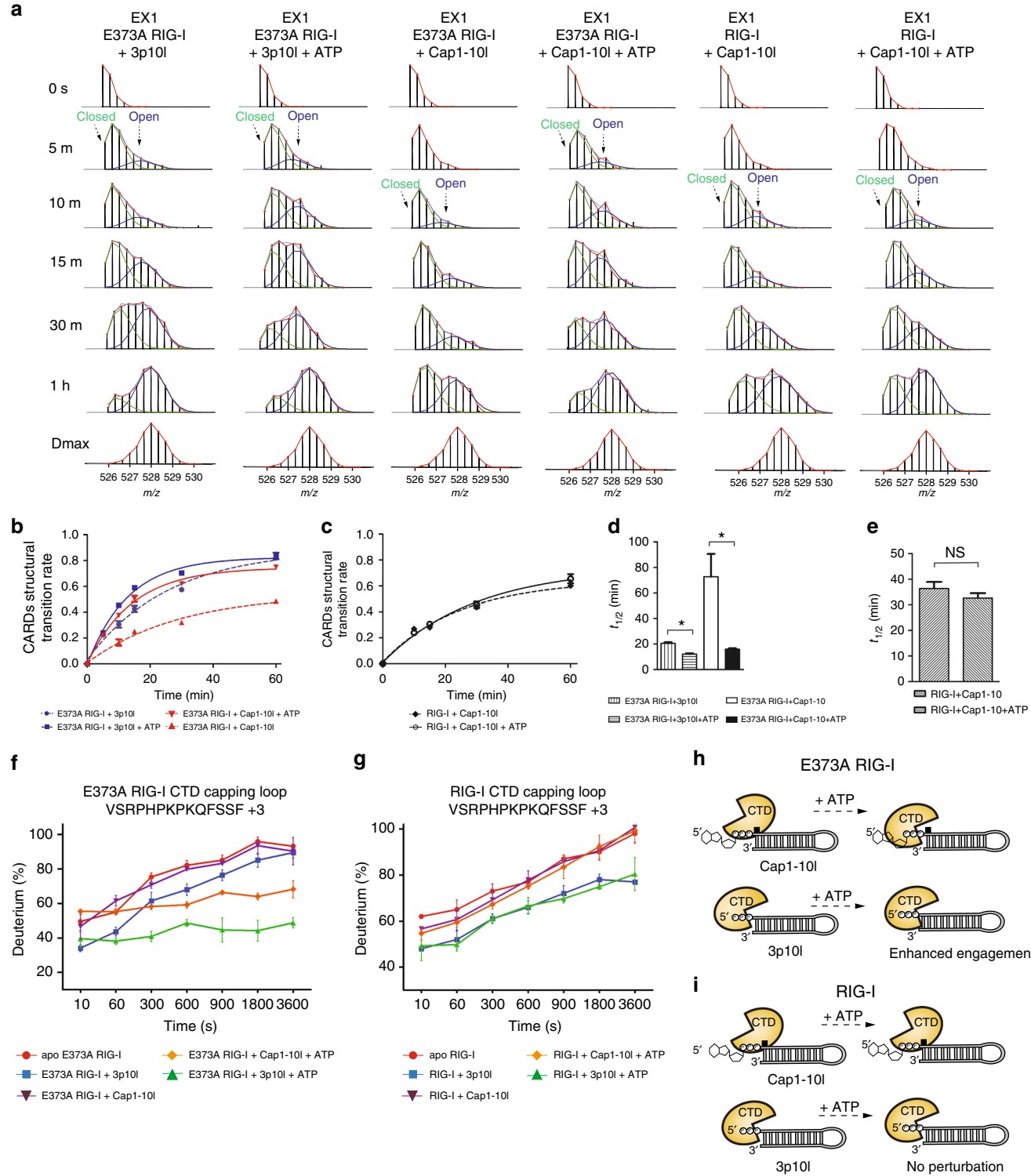

tethered in an ATP-bound state (Supplementary Fig. 1a, b and c, column (ix), (xi), (xx) and (xxii)).

Intriguingly, CARDs and HEL2i-gated regions of both RNA-RIG-I C268F complex displayed the greatest extent of deuterium uptake among all analyzed RNA-RIG-I combinations (Fig. 4a, b and Supplementary Fig. 1a, b and c, column (xxiv) and (xxv)). Analysis of EX1 kinetics further revealed a fast and similar transition rate of CARDs partial opening in both 3p10l-bound and Cap1-10l-bound C268F RIG-I complexes (9.8 ± 0.6 vs. 10.6 ± 0.6 min) regardless of differentially modified moieties at RNA

hairpin 5′ terminus (Fig. 4d, e and Supplementary Fig. 2g). Unlike WT/E373A RIG-I counterparts, CARDs of RNA-RIG-I C268F complexes were insensitive to external ATP or ADP.AlFx stimulus, whereas HEL-CTD module was sensitive to ADP.AlFx but not ATP binding (Supplementary Fig. 1a, b and c, column (xxvii), (xxviii), (xxix) and (xxx)). These results suggest that binding of self-RNA to C268F RIG-I results in CARDs activation even in the absence of ATP binding and hydrolysis.

We further tested 3p8l action, as it was unable to dislodge CARDs from its docking site on HEL2i in either WT or E373A

**Fig. 3** E373A affects RIG-I proofreading in an ATP-dependent manner. **a** MS spectra of WT and E373A RIG-I CARD2 latch peptide Y103–114 derived from indicated complexes in indicated on-exchange time points. The abundance of each mass population (high and low) is determined as Fig. 2a. **b**, **c** In each indicated state, the fraction of E373A (**b**) or WT (**c**) RIG-I CARDs molecules in the higher MS population (open conformation) to the total population is plotted against the on-exchange time points as Fig. 2c. **d**, **e** Half-life ($t_{1/2}$) of respective partial opening event is determined by fitting an exponential curve as Fig. 2d. (*Means that Cap1-10l-treated E373A RIG-I (20.51 min) and 3p10l-treated E373A RIG-I predicted $t_{1/2}$ (72.7 min) was higher than the upper limit (58.44 and 16.95, respectively, $\alpha = 0.05$), whereas $t_{1/2}$ calculated 3p10l- and ATP-treated E373A RIG-I (12.11 min) exceeded lower limit (13.33, $\alpha = 0.05$). Cap1-10l- and ATP-treated E373A did not exceed the lower or upper limits in this comparison. NS means statistically nonsignificant between compared groups. $t_{1/2}$ calculated for Cap1-10l-bound WT RIG-I and Cap1-10l- and ATP-treated WT RIG-I did not exceed the lower or upper limits ($\alpha = 0.05$) in this comparison.) **f**, **g** Differential deuterium uptake plots of CTD capping loop peptide region (VSRPHPKPKQFSSF, +3) of E373A (**f**) or WT RIG-I (**g**) upon receptor perturbed by 3p10l or Cap1-10l RNA in the presence or absence of ATP. The data are plotted as percent deuterium uptake vs. time on a logarithmic scale. The HDX plots of this CTD capping loop peptide between indicated groups were statistically analyzed by HDX Workbench (Supplementary Fig. 1c)[41]. **h**, **i** Schematic representations of CTD capping loop conformation associated with E373A (**h**) and WT RIG-I (**i**) upon the receptor interaction with ATP in the presence of Cap1-10l and 3p10l RNAs

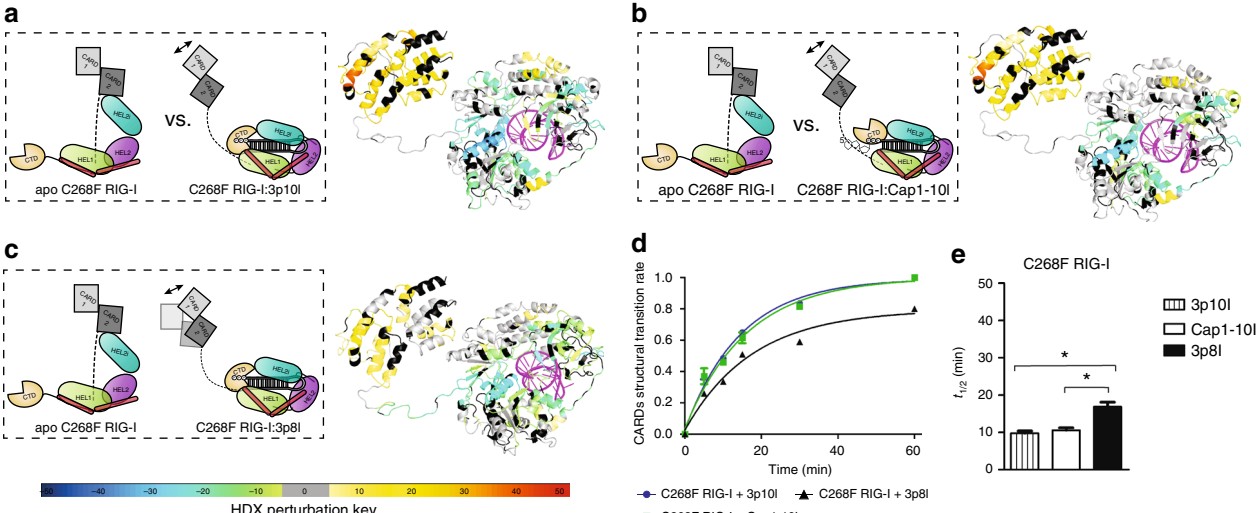

**Fig. 4** C268F RIG-I becomes constitutive signaling upon 5′ppp and Cap1 RNA binding. Schematic representations illustrate differential experiments of apo C268F RIG-I vs. C268F RIG-I associated with indicated RNA (on the left). Differential single amino acid consolidation HDX data are mapped onto the full-length RNA-bound RIG-I structure model in ribbon (on the right), as shown by representation of altered conformational dynamics of receptor upon binding to **a** 3p10l (Supplementary Fig. 1a, b and c, column (xxiv)), **b** Cap1-10l (Supplementary Fig. 1a, b and c, column (xxv)), and **c** 3p8l (Supplementary Fig. 1a, b and c, column (xxiv)). Percentages of deuterium differences are color coded according to the key (below). Black, regions that have no sequence coverage and include proline residue that has no amide hydrogen exchange activity; gray, no statistically significant changes between compared states; purple, duplex RNA ligand. **d** The fraction of C268F RIG-I CARDs molecules in the higher MS population (open conformation) to the total population is plotted against the incubation HDX time points as Fig. 2c. **e** Half-life ($t_{1/2}$) of the respective partial opening event is determined by fitting an exponential 3P with the prediction model as Fig. 2d. (* Means that $t_{1/2}$ calculated 3p8l-bound C268F RIG-I (16.85 min) exceeded the lower limit (13.80, $\alpha = 0.05$) in the indicated compassion, whereas both 3p10l- and Cap1-10l-bound C268F RIG-I did not exceed the lower or upper limits in this comparison

RIG-I (Supplementary Fig. 1a, b and c, column (vii) and (xviii)). Consistently with that observed with Cap1 and 3p10l, a similar HDX signature was observed in HEL-CTD with large buried areas involved in RNA stem binding and end capping of 3p8l (Supplementary Fig. 1a, b and c, column (xxvi)). CARDs of 3p8l-bound C268F RIG-I incorporated deuterium, but to a less extent than 10-bp hairpins, and became partially displaced into a solvent with a transition rate ($t_{1/2} = 16.8 \pm 1.3$ min) (Fig. 4c, d, e and Supplementary Fig. 2g). The recently solved X-ray structure suggests that this substitution in HEL1 motif I could displace K270 from its central position in the P-loop that coordinates interaction with ATP phosphate, into a position where K270 forms a salt bridge with E702[33,52]. This SMS mutation not only contributes to a tightly compressed HEL-CTD conformation that mimics ATP-bound state but also results in disruption of K270-mediated ATP hydrolysis activity. Thus, the hyperactive conformational flexibility of RIG-I C268F allows the receptor to become constitutively active in terms of signaling by displacing

the CARDs in a similar manner regardless of self or non-self-RNA interaction.

## Discussion

Autoimmune diseases such as Aicardi–Goutières and SMS are linked to gain-of-function RLR mutants with impaired ATPase activity[7,8,14,53,54]. So far, therapeutic interventions for SMS are hampered due to the lack of clarity on the mechanism driving their sustained IFN signaling and elevated inflammation. In this study, we attempt to uncover the initial activation events of dysregulated RIG-I proofreading associated with atypical SMS[7]. RIG-I activation is known to be coupled with the release of CARDs from an auto-inhibited conformation allowing it to form a complex with its adaptor protein MAVS[10,19,23,24,55]. Post-translational modifications, including ubiquitination and non-covalent K63-linked polyubiquitin chain, have been reported to be essential for RIG-I CARDs assembly[19,56,57]. A complex consisting of RIG-I tetrameric CARDs and four MAVS CARD

molecules is reported as a functional unit to nucleate MAVS CARD filament assembly[24]. This activated MAVS fiber further initiates downstream signaling cascades that result in upregulation of type I IFN and inflammatory cytokines[5,25–27].

Comprehensive differential HDX analysis of WT and SMS RIG-I complexes reveals that the initial events of RIG-I activation occur in a carefully choreographed order. Activation of RIG-I undergoes multiple checkpoints during RNA recognition, ATP binding, and ATP hydrolysis (Fig. 5a, b). Unlike PAMP RNA (e.g., Cap0, Cap0mA, and 5′ppp RNA hairpin), the presence of m7G cap and $N_1$-2′-O-methylation (Cap1) drastically compromises RNA-binding affinity to WT RIG-I and blunts CARDs destabilization nearly 3-fold (Fig. 5a). This undermined release of

CARDs may further correspond to delayed and reduced signal amplification downstream of the receptor. However, gain-of-function mutations enable the protein to restore such impairment by escaping these multiple checkpoints (Fig. 5b). Disruption of H830 that specifically coordinates the interaction with $N_1$-2′-O-methylation evades an early checkpoint and restores Cap1 RNA engagement for efficient CARDs dissociation similar to that observed for 5′ppp RNA interaction. Conformational flexibility of RIG-I enables different accommodation of 3p10l and 3p8l to HEL2i domain and activates its CARDs selectively.

ATP hydrolysis coordinates a more decisive and critical checkpoint to govern aberrant CARDs activation. Like WT, E373A RIG-I still enables discrimination against Cap1 via a H830

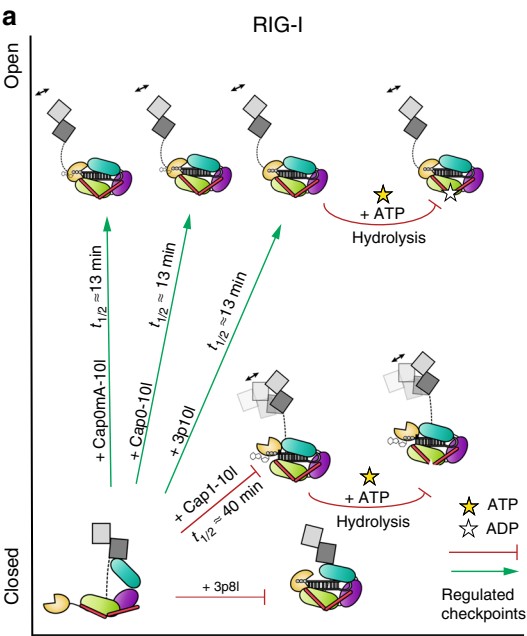

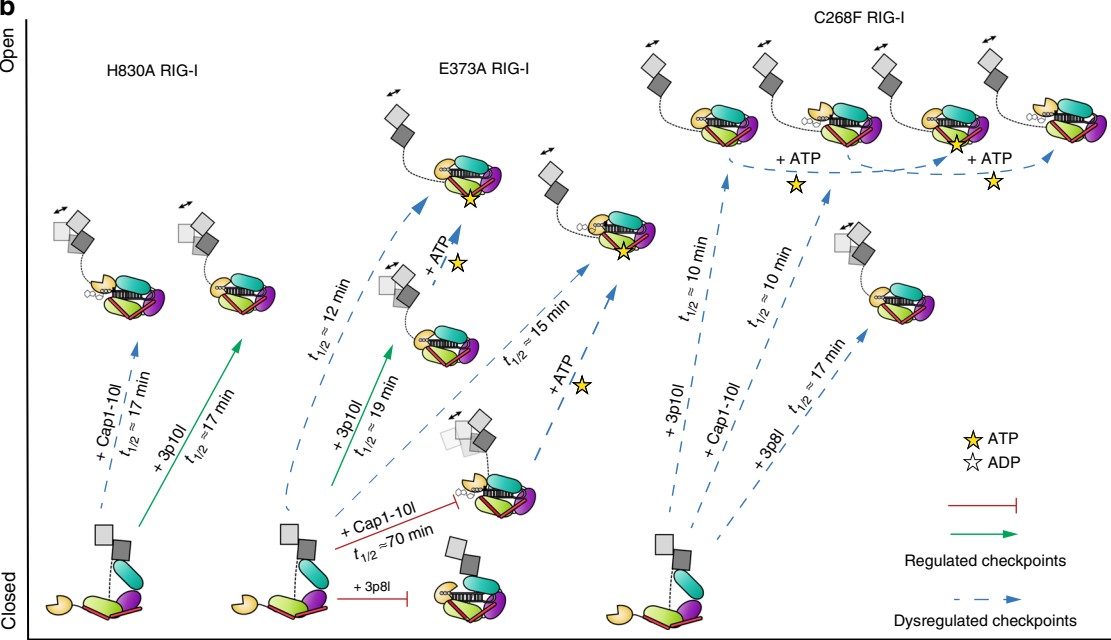

**Fig. 5** Functional and dysregulated checkpoints of RIG-I proofreading. **a** Schematic representations of functional RIG-I checkpoints governing RIG-I CARDs conformational transition. **b** Schematic representations of aberrant CARDs conformational transition of RIG-I gain-of-function mutants revealed by comprehensive differential HDX analysis with quantitative analysis of CARDs EX1 exchange kinetics

mechanism, resulting in slower CARDs partial opening than that for 5′ppp binding. To examine this further, protein-RNA complexes were incubated with 4 mM ATP, which is within the range of cellular ATP concentration. Dysfunctional ATP hydrolysis and ATP-binding energy have the protein-Cap1 complex transformed into a tightened HEL-CTD assemble mediated by RNA backbone-helicase engagement and tightened capping loop-Cap1 interaction. As a result, CARDs are sequentially released to an extent comparable to that of the 3p10l-E373A RIG-I-ATP complex. It is plausible that E373A substitution lowers affinity for $Mg^{2+}$ in the ATPase pocket and de-couples $Mg^{2+}$-ATP formation. E373A gain-of-function variant harnesses ATP molecule without hydrolyzing it for constitutive signaling.

In contrast, C268F substitution affords a more tethered HEL-CTD ring architecture regardless of 5′ end modifications. The recently reported crystal structure of the C268F complex reveals that F268 displaces K270 from its canonical position coordinating ATP γ-phosphate binding, into a position that forms a salt bridge with E702[52]. This amino acid rearrangement mimics the ATP-bound state in the absence of ATP and allows enhanced helicase engagement with both self and non-self RNAs. Interestingly, addition of ATP fails to further perturb the overall dynamics of RNA-bound C268F RIG-I complex. Combined, these observations suggest that the conformational flexibility of C268F RIG-I enables it to signal constitutively in an ATP-independent manner.

In summary, unrestrained RNA surveillance and dysregulated checkpoints coordinated by disordered RIG-I proofreading could occur through distinct mechanisms during different stages of the RIG-I signaling relay. This mechanism may apply to MDA5 as well as other innate immunity receptor-mediated autoimmune diseases. Confirmation of these mechanisms will assist develop therapies for the treatment of autoimmunity that drive immune defense against viral infections while minimizing autoimmune disorders.

## Methods

**RNA preparation**. All RNAs used in the study were chemically synthesized, high-performance liquid chromatography purified, and analyzed by mass spectrometry (Trilink and BioSynthesis RNA synthesis service). The lyophilized RNA was resuspended in 20 mM potassium phosphate buffer, pH 7.0.

**Protein preparation**. Human WT full-length RIG-I (1–925) was cloned and expressed in pET28 SUMO vector. Full-length RIG-I mutants H830A, E373A, and C268F, was constructed by site-directed mutagenesis of full-length RIG-I in pET28 SUMO vector with QuikChange XL Site-Directed Mutagenesis Kit (Agilent Technologies) following the protocol with corresponding primers (Supplementary Table 1). Constructs were confirmed by sequencing. Human full-length RIG-I (1–925) and mutants were expressed in *Escherichia coli* strain Rosetta (DE3) (Novagen) as soluble proteins. The soluble fraction of protein was purified from the cell lysate using a HisTrap FF column (GE Healthcare). The recovered protein was then digested with Ulp1 protease to remove the 6 × His-SUMO tag and further purified by hydroxyapatite column (CHT-II, Bio-Rad) and heparin-sepharose column (GE Healthcare). Finally, purified protein was dialyzed against 50 mM HEPES (pH 7.5), 50 mM NaCl, 5 mM MgCl₂, 5 mM dithiothreitol (DTT), 10% glycerol overnight at 4 °C, snap frozen in liquid nitrogen, and stored at −80 °C. Human RIG-I CARDs (1–228) was cloned in pGEX 6P-1 GST vector. RIG-I CARDs was expressed in *Escherichia coli* strain Rosetta (DE3) (Novagen) as soluble proteins. The soluble fraction of protein was purified from the cell lysate using a GST affinity column (GE Healthcare). The recovered protein was then digested with PreScission protease to remove the GST tag and further purified by HiTrap FF Q column (GE Healthcare). Finally, purified protein was passed through Superdex200 10/300 Increase size exclusion column against 25 mM Tris, pH 8.0, 50 mM NaCl, 5 mM DTT, 10% glycerol overnight at 4 °C, snap frozen in liquid nitrogen, and stored at −80 °C.

**Hydrogen/deuterium exchange mass spectrometry**. *Peptide identification*: Peptides were identified using tandem MS (MS/MS) with an Orbitrap mass spectrometer (Q Exactive, Thermo Fisher). Product ion spectra were acquired in data-dependent mode with the top five most abundant ions selected for the product ion analysis per scan event. The MS/MS data files were submitted to Mascot (Matrix Science) for peptide identification. Peptides included in the HDX analysis peptide set had a MASCOT score >20, and the MS/MS spectra were verified by

manual inspection. The MASCOT search was repeated against a decoy (reverse) sequence and ambiguous identifications were ruled out and not included in the HDX peptide set.

*HDX-MS analysis*: Ten micromoles of WT/H830A/E373A/C268F RIG-I receptors or CARDs (50 mM HEPES, pH 7.4, 150 mM NaCl, 5% glycerol, 5 mM MgCl₂, 2 mM DTT) were incubated with the respective RNA ligands (Supplementary Fig. 1a) at a 1:1.2 molar ratio for 1 h (protein-ligand) before the HDX reactions at 4 °C. Indicated protein-RNA complexes (Supplementary Fig. 1a) was further incubated with 4 mM ATP (Sigma) or 4 mM ADP.AlFx for 1 h to reach to conformational equilibrium states. Five microliters of protein-protein complex with ligand/peptide was diluted into 20 μL D₂O (deuterium) on exchange buffer (50 mM HEPES, pH 7.4, 150 mM NaCl, 5 mM MgCl₂, 2 mM DTT) and incubated for various HDX time points (e.g., 0, 30, 60, 300, 600, 900, 1800, and 3600 s) at 4 °C and quenched by mixing with 25 μL of ice-cold 4 M guanidine hydrochloride and 1% trifluoroacetic acid. Dmax samples were incubated in D₂O on exchange buffer containing 3 M guanidine hydrochloride (50 mM HEPES, pH 7.4, 150 mM NaCl, 5 mM MgCl₂, 2 mM DTT, and 3 M guanidine hydrochloride) overnight at room temperature. The sample tubes were immediately placed on dry ice after the quenching reactions until the samples were injected into the HDX platform. Upon injection, samples were passed through an immobilized pepsin column (2 mm × 2 cm) at 200 μL/min, and the digested peptides were captured on a 2 mm × 1 cm C₈ trap column (Agilent) and desalted. Peptides were separated across a 2.1 mm × 5 cm C₁₈ column (1.9 μm Hypersil Gold, Thermo Fisher) with a linear gradient of 4–40% CH₃CN and 0.3% formic acid, over 5 min. Sample handling, protein digestion, and peptide separation were conducted at 4 °C. Mass spectrometric data were acquired using an Orbitrap mass spectrometer (Q Exactive, Thermo Fisher) with a measured resolving power of 65,000 at $m/z$ 400. HDX analyses were performed duplicate or triplicate, with single preparations of each protein ligand complex. The intensity weighted mean $m/z$ centroid value of each peptide envelope was calculated and subsequently converted into a percentage of deuterium incorporation. In the absence of a fully deuterated control, corrections for back-exchange were made by an estimated 70% deuterium recovery, and accounting for the known 80% deuterium content of the deuterium exchange buffer. When comparing the two samples, the perturbation %D is determined by calculating the difference between the two samples. HDX Workbench colors each peptide according to the smooth color gradient HDX perturbation key (D%) shown in each indicated figure. Differences in %D between −5 to 5% are considered non-significant and are colored gray according to the HDX perturbation key[41]. In addition, unpaired $t$ tests were calculated to detect statistically significant ($p < 0.05$) differences between samples at each time point. At least one time point with a $p$ value <0.05 was present for each peptide in the data set further confirming that the difference was significant.

*Data rendering*: The HDX data from all overlapping peptides were consolidated to individual amino acid values using a residue averaging approach. Briefly, for each residue, the deuterium incorporation values and peptide lengths from all overlapping peptides were assembled. A weighting function was applied in which shorter peptides were weighted more heavily, and longer peptides were weighted less. Each of the weighted deuterium incorporation values was then averaged to produce a single value for each amino acid. The initial two residues of each peptide, as well as prolines, were omitted from the calculations[40].

**EX1 kinetics analysis and curve fitting**. The isotopic distributions of MS peak (Y101–114, +3) is specifically analyzed by the HX express2 software[58,59]. Exponential 3P with the prediction model: $a + b.\exp(c.\text{time}(\min)$, where $a$ was the asymptote, $b$ was the scale, and $c$ was the growth rate, was used to fit a curve to %D (response) and time (regressor) for each compound state. The inverse prediction was used to solve for the half-life ($t_{1/2}$) for each compound state. Analysis of means was performed to identify half-life ($t_{1/2}$) of compound states that were statistically different from the average half-life ($t_{1/2}$) in this given comparison group. JMP®, Version 13.2.1. SAS Institute Inc. (Cary, NC, USA), 1989-2007 was used to perform the curve fit, and α was set to 0.05. Predicted half-life values ($t_{1/2}$) that exceed lower or upper limits were considered statistically significant.

**BIOMOL Green ATPase assay**. Five picomoles of Cap1-10l or 3p10l was, respectively, incubated with WT/E373A/C268F RIG-I receptors in a total of 50 μL ATPase buffer (50 mM HEPES, pH 7.4, 150 mM NaCl, 5 mM MgCl₂, 2 mM DTT) for 30 min. ATP (Sigma) was added to the reaction mixture at a final concentration of 2 mM, and the reactions were incubated at room temperature for 5 to 10 min. Free phosphate concentration was determined using BIOMOL Green reagent (Enzo Life Sciences) in a microplate format, and absorbance was measured at OD₆₃₀ₙₘ. The significance of differences between groups was evaluated by unpaired Student's $t$ test (*$p < 0.05$).

**IFN Dual Reporter Assays**. hRIG-I (accession number NM_014314.3, NP_055129.2)-pUNO vector (Invivogen, Cat# puno1-hRIG-I) was mutated using NEB Q5 Site-Directed Mutagenesis Kit (Cat# E0554S). Mutation (C268F, K270A, E373A, and H830A) primers were designed with the NEBaseChanger software (NEBaseChanger.neb.com) and were diluted to 10 μM. pUNO-hRIG-I vector was

diluted to 10 ng/μL. A 25 μL reaction was prepared by mixing 9.0 μL of nuclease-free water, 1 μL of template DNA, 1.25 μL of forward primer, 1.25 μL of reverse primer, and 12.5 μL of Q5 Hot Start High-Fidelity 2X Master Mix. Cycling conditions were set to an initial denaturation at 98 °C for 30 s, then 25 cycles (10 s at 98 °C, 20 s at Ta provided by the NEBaseChanger software, 90 s at 72 °C) and a final extension for 2 min at 72 °C. All mutagenesis reactions were further subject to the NEB KLD reaction where 1 μL of the 25 μL PCR reaction was mixed with 5 μL of 2× KLD reaction buffer, 1 μL of 10× KLD Enzyme Mix, and 3 μL of nuclease-free Water. The reaction was incubated at room temperature for 5 min. Five microliters of the KLD mix were added to 50ed using internal sequencing primers. A total of $2.5 \times 10^5$ HEK293T cells (ATCC CRL-3216) were co-transfected with 1 μg IFN-β-luc plasmids, 0.5 μg TK-Renilla-luc plasmid, and 1.5 μg WT pUNO-RIG-I or RIG-I mutants (C268F RIG-I, E373A RIG-I, or H830A). After 24 h, each transfectant groups were transferred into 96-well plates ($4.5 \times 10^4$ cells/well). After 4 h, each ds 100 ng RNAs per well (3p8l, 3p10l, or Cap1-10l) was transfected using Lipofectamine 2000 (Invitrogen). After 24 h, dual luciferase activity was measured using the Dual-Glo luciferase assay system (Promega). The luciferase activity was determined with Perkin-Elmer EnVision Multilabel Reader. The significance of differences between groups was evaluated by unpaired Student's $t$ test ($*p < 0.05$; $**p < 0.01$).

**RNA binding and ATP hydrolysis**. The ATP hydrolysis activity was measured at constant RIG-I (5 nM) and increasing RNA concentration (1 nM–1 μM) in the presence of 1 mM ATP (spiked with [γ-$^{32}$P]ATP). A time course (0–60 min) of the ATPase reactions was performed in buffer A (50 mM MOPS, pH 7.4, 5 mM DTT, 5 mM MgCl$_2$, 0.01% Tween-20) at 37 °C. The reactions were stopped at desired time points using 4 N formic acid and analyzed by PEI-Cellulose-F TLC (thin-layer chromatography) (Merck) developed in 0.4 M potassium phosphate buffer (pH 3.4). The TLC plates were exposed to a phosphor-imager plate, imaged on a Typhoon phosphor-imager, and quantified using the ImageQuant software. The ATPase rate was determined from the plots of [Pi] produced vs. time and dividing the rate of hydrolysis by RIG-I concentration. The ATPase rates were then plotted as a function of RNA concentration and fitted to hyperbolic equation (Eq. 1) or quadratic equation (Eq. 2) to get the binding affinity ($K_{d,app}$) and the maximal ATPase rate ($k_{atpase}$), where observed ATPase rate = $k_{atpase}$ × [PR]/[Pt], where [Pt] is the total protein concentration, [PR] is the amount of RIG-I-RNA complex formed, and [R] is the RNA concentration being titrated

$$[\text{PR}] = \frac{[R]}{K_{d,app+[R]}}, \tag{1}$$

$$[\text{PR}] = \frac{\left([P_t] + [R_t] + K_{d,app}\right) - \sqrt{\left([P_t] + [R_t] + K_{d,app}\right)^2 - 4[P_t][R_t]}}{2}. \tag{2}$$

## Data availability

Further data supporting the findings of this study are available from the corresponding author upon reasonable request.

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

## Acknowledgements

This research was supported in part by the National Institutes of Health, National Institute for General Medicine Grants GM111959 (S.S.P. and J.M.) and GM103368 (J.M. and P.R.G.), and the Intramural Research Program of the National Institute of Allergy and Infectious Diseases/National Institutes of Health (J.M.).

## Author contributions

J.Z., S.S.P., J.M., and P.R.G. designed the experiments. S.C.D. and R.D.G-O. made the mutants, S.C.D. and performed the RNA-binding and ATPase studies, and B.S. and C.W. purified the mutants. S.S.P., S.C.D., B.S. designed the biochemical experiments. M.R.C. performed interferon assays. G.C.C. and B.D.P. assisted HDX data analysis. J.Z., S.J.N., and P.R.G. performed HDX experiments. J.Z., C.W., S.S.P., J.M., and P.R.G. wrote the manuscript and accommodated comments from all authors.

## Additional information

**Competing interests:** The authors declare no competing interests.

