## [Peer Review File · Nature Communications]

Reviewers' comments:

Reviewer #1 (Remarks to the Author):

The authors describe their study of RIG-I function in the different steps of RNA recognition and response. The model here is to assess purified RIG-I at different stages of RNA binding, CARD release from autorepression by the CTD, ATP hydrolysis, and ultimate signaling via a luciferase IFN-beta promoter reporter assay. The approach is innovated to apply a combination of HSX-MS and a level of functional assays to determine the effect from E373A and C268F mutations on the different actions of RIG-I in interacting with nonself (PAMP) and self RNA. The previous work from this group showed that H830 imparts restriction of 2'O methyl RNA (cap1-2 RNA) to prevent high affinity binding of self RNA. The current now assesses how RIG-I mutations associated with specific autoimmune disorders in humans permits self RNA binding and RIG-I activity binding despite the H830 restriction. They show that ATP binding by the E373A mutant of RIG-I places RIG-I in a ATP-on state to release the CARDS for signaling- shown by induction of IFN-beta luciferase and RIG-I structure analysis. But C268F simply places the CARDS in a conformation displaced from autorepression and now in a signaling-on state independent of ATPase. The study validates the checkpoint control of RIG-I operating through 1) RN binding affected by RNA modification (methylation/cap/5'ppp), 2) ATP occupancy and hydrolysis, and 3) CARD intramolecular interactions within the Hel and CTD to control RIG-I off-on signaling conformation. The new data here is the really the careful examination of the E373 and the C268 aa sites in the regulation of RIG-I and linkage with the autoimmune conditions of Singleton-Merten Syndrome.

Specific comments:

The structural studies using HDX-MS are important to reveal distinct conformation changes of RIG-I directed by the different tester RNA species in vitro. However, the study lacks any ex vivo or in vivo validation. Can RIG-I really bind to self RNAs under the conditions within a living cell or human in the context of WT vs E373 vs C268 mutation? No evidence for these interactions are shown beyond the highly artificial tester RNAs shown in Figure 1. The authors need to conduct expression and pull-down of RIG-I and RIG-I mutants from cultured cells at least, and then assess bound RNAs using typical recovery and sequencing approaches. If their model is correct, then the mutant vs WT RIG-I should be associated with self RNAs.

In terms of RIG-I signaling the authors show IFN-beta luc assay data but this data set needs validation beyond a simple luc assay. Analysis of ISG mRNA or protein expression (for example IFIT, IFIT2, MX, OAS..) should be conducted to verify induction of the cellular response to RIG-I signaling and IFN induction (also a measure of actual IFN production levels using an IFN ELISA assay is typically used here to validate IFN-beta luc results).

Do the RIG-I mutants differentially impact IRF3 and NF-kB activation? The authors need to assess IRF3 phosphorylation and NF-kB activation directly. In light that the different mutants redirect RIG-I checkpoint signaling actions, the impact on these critical downstream transcription factors should be addressed. It is possible that the E373 vs C268 could impact IRF3 activation kinetics but not NFkB, for example, by placing RIG-I in different conformations for differential interaction with downstream signaling partners that impact IRF3 or NFkB signaling.

Do the different RIG-I mutants form a stable complex with MAVS? This component of the analyses should have been included but is not even mentioned nor discussed. When activated RIG-I will form a stable complex with the MAVS adaptor protein. This complex is essential for the induction of IFN and the actions of RIG-I. Thus, one should expect that the E373 and C268F mutants would form a constitutive complex with MAVS or a complex that is enhanced by RNA. How does each mutant impact MAVS binding, and what are the differential kinetics of this interaction in the presence or absence of the different tester RNAs?

What is the impact of each RIG-I mutant in the context of virus infection where RIG-I is expected

to mediate virus recognition. Do the mutants compromise this activity or do they confer enhanced resistance against infection?

Statistics should be included for all data sets applicable. Please include p values and statistical methods used for the data and differences shown.

Discussion: this section needs to include a broader presentation of how RIG-I checkpoints impact the entire signaling cascade from RNA binding, RIG-I activation, MAVS interaction, IRF activation, and gene expression.

Reviewer #2 (Remarks to the Author):

The manuscript by Zheng et al investigates the molecular details of how the Retinoic acid Inducible Gene-I (RIG-I) Receptor interacts with different RNAs and how mutations in RIG-I cause dysregulation. The role of RIG-I and other major intracellular immune receptors in autoimmune disease is important and more information is needed for how they activate and discriminate between viral RNA species and self RNA – and how mutations result in dysregulation and disease. In this work, the authors extend prior work that also made extensive use of the HDX-MS method, in which they studied the apo state of RIG-I and its binding to ATP, triphosphorylated RNA and longer duplex RNA (Zheng J, Nucleic Acid Research. 2015). That prior work provided a model for how RIG-I (and a related receptor) in the autoinhibited apo-state undergoes allosteric changes when transitioning to an active state.

The present manuscript builds on this work and studies the binding of RIG-I (and mutants H830A, C268F and E373A) to different RNAs. The work provides new mechanistic insights into RIG-I regulation. Most interestingly, the results provide a model of how the specific gain-of-function mutations C268F and E373A in RIG-I cause dysregulation leading to Singleton-Merten syndrome, an autoimmune disease.

The experimental work, in particular the HDX-MS component, is very extensive and well-performed. Furthermore, the conclusions of the work are interesting and impactful, also considering that the proposed mechanisms of RIG-I dysregulation by mutations could be relevant for other innate immunity receptors. I am positive towards publication. I have some specific concerns, in particular about the conclusions drawn from the EX1 kinetic data, that need to be addressed prior to publication.

Major comments:

Concerning the observed EX1 kinetics:

Firstly, there are a few sentences detailing EX1 theory that needs to be revised/clarified:

Line 203: The authors write: The CARD2 latch region (Y101-114), which is spatially locked to the HEL2i gate motif in the apoform, displays EX1 exchange behavior upon binding to PAMP RNA, suggesting this region of the receptor is conformationally heterogeneous in solution (4). I think this statement is unclear. The mere presence of EX1 does not show that the receptor conformationally heterogeneous. EX1 informs on the timescale at which structural transitions that facilitate exchange occurs in the protein. A protein undergoing EX2 could be equally conformationally heterogeneous.

Line 204: The authors write: "If an unfolding event occurs slow enough for the backbone amide hydrogens within the unfolding region to be fully exchanged, EX1 kinetics is observed with bimodal distribution". It is when the rate of the refolding or closing event (k_{cl}) is sufficiently slow that EX1

kinetics is observed – please correct the sentence accordingly.

Line 210: The authors write: “Therefore, the region undergoing EX1 kinetics is a mixture of unfolded and folded conformers, which transitions from one state to the other via a slow and correlated exchange event.”. This sentence is misleading, the transition is slow – and thus the exchange is correlated. Please rephrase.

Secondly, the authors should be more specific concerning what they can conclude and what they cannot solely based on the presence of EX1 kinetics observed for RIC-I. To me the authors lack functional evidence to support that the observed EX1 kinetics observed (and the folded and unfolded states involved) correspond to the inactive and active forms of RIG-I. I agree that it seems enticing to think so (and very interesting) but that does not necessarily make it so. Do the authors have functional data that show that the derived $t_{1/2}$ values are on the same timescale as the rate of activation or enzymatic activity of RIG-I. If so, such a comparison would make their ultimate conclusions from the EX1 observations much more interesting and convincing.

For instance:

Line 232: Due to the induced appearance of EX1 kinetics upon RNA binding the authors write “RNA binding by RIG-I drives CARDS module from a closed conformation to a partially opened conformation (Fig. 2a)”.

Line 238: To more precisely measure the increased correlated exchange, we determined the CARDS transition rate from the inactive to the active state”

The presence of EX1 kinetics and the derived $t_{1/2}$ values shows that the rate of unfolding and refolding is slow. But the authors cannot strictly speaking conclude that the slow unfolding/refolding transition observed in the CARDS module is a transition from an inactive to an active state. Additional functional data to support this conclusion is needed, for instance as described above. Alternatively, this caveat needs to be clearly underlined.

Finally on that note, the authors should investigate further if the auto-inhibited apo RIG-I and apo C268F, undergo EX1 kinetics by probing longer timescales (see comment below concerning Fig. 2a). If they do, then one could suppose that these two forms of RIG-I should have some low residual activity, due to the slow build-up of the supposed open and active state, according to the theory put forward by the authors.

Figure 2a, column 1 + line 221-223. “The EX1 kinetic regime of CARD2 latch peptide in apo RIG-I was difficult to detect as there is very little solvent exchange for the auto-inhibited domain” The Apo state exchanges so little that the presence or absence of EX1 kinetic cannot be determined based on the current data. This can likely be examined by prolonging the time course or maybe increasing the temperature. The current experiments are performed at 4°C. Also, it would appear that an unfolded population occurs at 1 hr for wt RIG-I bound to 3p8I, but not so for apo RIG– thus repeating the exchange experiment at longer times for both state would be informative and could reveal an effect of 3p8I. Finally, it is not clear to me why the apo-RIG is not simply referred to as wt RIG-I in the figure.

Line number 148-155. A reduction on 12 % is mentioned. It is not defined whether it is significant or not. The methods section does not describe how significance is determined - this is defined in the SI, but, in my opinion, this is an important parameter that should be easily accessible in the manuscript and when reviewing the data. Especially, when as all structural discussions are based on the HDX-MS data.

Minor comments:

Abstract: Several sentences in the Abstract needs are very long and unclear and must be revised.

For instance: "A RIG-I residue (H830) mediates specific sensing of 5' 7-methyl guanosine and 2'O-methylated on the first base (Cap1) self RNA and is coupled with a threefold delay in Caspase Activation and Recruitment Domains (CARDs) partial unfolding event compared to that of 5'ppp RNA".

Line 172 and throughout: the authors use the term "solvent exchange". To me this is unspecific and strictly speaking incorrect as solvent per se is not exchanged. It could thus confuse a non-expert – I recommend use of a more specific term like HDX, hydrogen exchange or deuterium uptake etc.

Figure 1b. Please highlight/bold the R1 and R2 to guide the reader.

Figure 1c. Suggestion – add the origin of the different RNA constructs. e.g. Self-RNA or Viral RNA to the figure. This will make it easier for readers not familiar with the research area to follow the argumentation.

Line number 504. Missing letter in faction, should be fraction.

Line number 148-150. "The HDX data obtained for sequence overlapping peptides were consolidated to individual amino acid values using a residue averaging approach (29)". For people without former knowledge of HDX, it could sound like they have residue-resolved HDX-data everywhere. This is probably not the case. Please elaborate. Also, for all overlapping peptides for which this procedure was applied, the authors should inspect the maximal-labeled control and verify that the a similar back-exchange was observed. The Schriemer lab has observed that overlapping peptides can exhibit large differences in back-exchange rendering subtractive analysis problematic for those peptides.

Figure 2a, column 2 + line 223-226: "In contrast, CARDs (1-228) protein, where the CARDs domain is not auto-inhibited and fully exposed to solvent, resulted in rapid deuterium incorporation and the latch peptide underwent EX2 exchange indicating structural homogeneity in solution". To me, this does not look like pure EX2 kinetics. The peak width at 10min and 15min is markedly larger than at 30min and 1h. Hinting at a possible mixture of kinetics (EXX). Furthermore, the apo CARD state is defined as "open", but its conformation is not similar to the "open" state described with blue in columns 4-9, as it takes the apo CARDs around 30min to be fully deuterated.

Line 439-440. "HDX MS data was calculated with the in-house developed software and corrected for back-exchange on an estimated 70% recovery."

Figure 2a shows that the authors have recorded 'maximally labelled' samples. Did they normalize the deuterium incorporation to these peptide specific values or to the average back-exchange reported to be 30%. I would strongly recommend the former. Please elaborate on the procedure used and why.

Figure 3.d-g Non-linear or logarithmic x-axis on graph, d-g. Please remove the s in the single time point labels on the x-axis.

Supporting Information: The HDX-MS experimental section has some cases of incorrect nomenclature, spellings and unexplained abbreviations etc. For instance, gHCL, D20 etc. Please revise carefully and consistently.

Response to reviewers: Point-by-Point: **Author's responses are in red.**

Reviewers' comments:

Reviewer #1 (Remarks to the Author):

The authors describe their study of RIG-I function in the different steps of RNA recognition and response. The model here is to assess purified RIG-I at different stages of RNA binding, CARD release from autorepression by the CTD, ATP hydrolysis, and ultimate signaling via a luciferase IFN-beta promoter reporter assay. The approach is innovated to apply a combination of HSX-MS and a level of functional assays to determine the effect from E373A and C268F mutations on the different actions of RIG-I in interacting with non-self (PAMP) and self RNA. The previous work from this group showed that H830 imparts restriction of 2'0 methyl RNA (cap1-2 RNA) to prevent high affinity binding of self RNA. The current now assesses how RIG-I mutations associated with specific autoimmune disorders in humans permits self RNA binding and RIG-I activity binding despite the H830 restriction. They show that ATP binding by the E373A mutant of RIG-I places RIG-I in a ATP-on state to release the CARDS for signaling- shown by induction of IFN-beta luciferase and RIG-I structure analysis. But C268F simply places the CARDS in a conformation displaced from autorepression and now in a signaling-on state independent of ATPase. The study validates the checkpoint control of RIG-I operating through 1) RN binding affected by RNA modification (methylation/cap/5'ppp), 2) ATP occupancy and hydrolysis, and 3) CARD intramolecular interactions within the Hel and CTD to control RIG-I off-on signaling conformation. The new data here is the really the careful examination of the E373 and the C268 aa sites in the regulation of RIG-I and linkage with the autoimmune conditions of Singleton-Merten Syndrome.

Specific comments:

1. The structural studies using HDX-MS are important to reveal distinct conformation changes of RIG-I directed by the different tester RNA species in vitro. However, the study lacks any ex vivo or in vivo validation. Can RIG-I really bind to self RNAs under the conditions within a living cell or human in the context of WT vs E373 vs C268 mutation? No evidence for these interactions are shown beyond the highly artificial tester RNAs shown in Figure 1. The authors need to conduct expression and pull-down of RIG-I and RIG-I mutants from cultured cells at least, and

then assess bound RNAs using typical recovery and sequencing approaches. If their model is correct, then the mutant vs WT RIG-I should be associated with self RNAs.

Response to comment 1:

We apologize for our oversight by excluding the wealth of information already available on self RNAs binding by RIG-I. We have revised the manuscript and included the necessary information related to the self RNAs in the introduction section.

Specifically, this question concerns whether wild-type or mutant RIG-I can actually bind to self-RNAs in living cells or *in vivo*. Several groups have published the relevant findings and have proven the binding of RIG-I to self-RNAs found *in cellular* [1, 12] and *in vivo* [2]. For instance, one study has performed RIG-I RNA immunoprecipitation from cell lysates of the murine splenic B-cell line and provided direct evidence that WT RIG-I recognizes several regions within NF- κ B1 3' -UTR mRNA [12]. Lässig *et al.* have performed immunoprecipitation of RIG-I-RNA complex from virus infected and non-infected HEK293T RIG-I KO cells, which showed the interaction of RIG-I ATPase deficient mutant E373Q with host ribosomal RNAs. Similarly, an increased amount of RNA is reported to be co-purified from C268F and E373A RIG-I from uninfected cells compared to that of WT RIG-I [1]. Schuberth-Wagner *et al.* showed that cellular RNAs can activate H830A mutant RIG-I but not WT RIG-I [7]. Another published study using an *in vivo* murine model validated that small self-RNA fragments generated by RNase L can trigger IFN β responses via the RIG-I signaling pathway [2]. Taken all together, these published studies address the question as to whether wild-type or mutant RIG-I can bind to self-RNAs *in cellular* or *in vivo*.

It is important to note that the RNAs selected in this study have been well-documented in various published articles and are considered as representative of validated RNA ligands for RIG-I. As such, these RNAs are optimal *in vitro* ligands [3-7]. The focus of the current study is to use these functionally validated RNA ligands to probe RIG-I-RNA interactions by HDX and gain insight into the structural mechanism of receptor activation. The observations from these biophysical studies are then correlated with findings in previously described studies focused on dysregulation of RIG-I in autoimmune disease.

2. In terms of RIG-I signaling the authors show IFN-beta luc assay data but this data set needs validation beyond a simple luc assay. Analysis of ISG mRNA or protein expression (for example

IFIT, IFIT2, MX, OAS..) should be conducted to verify induction of the cellular response to RIG-I signaling and IFN induction (also a measure of actual IFN production levels using an IFN ELISA assay is typically used here to validate IFN-beta luc results).

Response to comment 2:

Several published studies have shown that IFN assays correlate with respective ISG mRNA expression level as well as protein levels [13, 14]. For instance, one study has shown that high level of ISG15 gene expression in E373A and C268F RIG-I transfected HEK293T cells (non-infected) correlates with strong *IFNB1* gene expression [13]. One recent review article also states that over-expression of mutant RIG-Is (E373A and C268F) sufficiently induced IRF3 phosphorylation and IRF3 dimerization. As a result, RIG-I mutations led to increased expression of IFN- β and ISG15 in untreated cells as well as in polyI:C transfected cells [10]. We have revised the manuscript and added this information in the introduction section.

Do the RIG-I mutants differentially impact IRF3 and NF-kB activation? The authors need to assess IRF3 phosphorylation and NF-kB activation directly. In light that the different mutants redirect RIG-I checkpoint signaling actions, the impact on these critical downstream transcription factors should be addressed. It is possible that the E373 vs C268 could impact IRF3 activation kinetics but not NFkB, for example, by placing RIG-I in different conformations for differential interaction with downstream signaling partners that impact IRF3 or NFkB signaling.

Response to comment 3:

As stated in response 2, one published study has already shown that RIG-I SMS mutants E373A and C268F similarly impact IRF3 phosphorylation activity and NF-kB activation in the non-infected HEK293T cells [13]. The impact on RIG-I downstream signaling pathways has been well described in the literature and supports the findings from our biophysical mechanistic study. Our current study presented here focuses on the direct biophysical analysis of RIG-I-RNA binding at the initial step of activation rather than related downstream signaling pathways. We have revised the manuscript and added this information in the introduction section.

Do the different RIG-I mutants form a stable complex with MAVS? This component of the analyses should have been included but is not even mentioned nor discussed. When activated RIG-I will form a stable complex with the MAVS adaptor protein. This complex is essential for the induction of IFN and the actions of RIG-I. Thus, one should expect that the E373 and C268F mutants would form a constitutive complex with MAVS or a complex that is enhanced by RNA. How does each mutant impact MAVS binding, and what are the differential kinetics of this interaction in the presence or absence of the different tester RNAs?

Response to comment 4:

There is ample literature showing RIG-I CARDs interact and nucleate MAVS filament formation [15-19]. A recent review article on RIG-I-like receptor states that “the importance of excess of RLR-dependent signaling via MAVS leading to IFN signature in the pathogenesis of these autoimmunity has been clarified” [9]. The barrier to RIG-I-MAVS CARD activation is the sequestration of CARDs by the Hel2i domain and involvement of K63-linked ubiquitin chains (K63-Ub_n). The Sun Hur group has solved the atomic structure of both the RIG-I CARDs tetramer attached with K63-Ub₂ and the RIG-I CARDs tetramer complexed with four MAVS CARD molecules [16, 19]. These studies clearly demonstrate that RIG-I CARDs, by forming a helical tetrameric structure, acts as a template for the MAVS CARD filament assembly.

Regardless, our current study focuses on the initial activation steps of wt and mutant RIG-I to provide direct conformational dynamic information on RNA recognition and discrimination, ATP binding, and ATP hydrolysis. The MAVS CARD filament activation involves the interaction between RIG-I CARDs tetramer and MAVS CARD as well as the high affinity binder of K63 poly-ubiquitin chains [16]. While studying the kinetics of RIG-I-MAVS interaction is very interesting, it is outside the focus of the current study we present here.

What is the impact of each RIG-I mutant in the context of virus infection where RIG-I is expected to mediate virus recognition. Do the mutants compromise this activity or do they confer enhanced resistance against infection?

Response to comment 5:

The RIG-I mutations E373A and C268F are implicated in autoimmunity and constitutive signaling of the receptor, rather than implicated in viral signaling, whereas the RIG-I mutant H830A has been reported to be able to recognize Cap1 RNAs found in viruses like yellow fever virus [7].

Statistics should be included for all data sets applicable. Please include p values and statistical methods used for the data and differences shown.

Response to comment 6:

We agree, and we have carefully addressed this comment in the revised manuscript showing all statistical data. Please see the revised manuscript figures and supplementary figures.

Discussion: this section needs to include a broader presentation of how RIG-I checkpoints impact the entire signaling cascade from RNA binding, RIG-I activation, MAVS interaction, IRF activation, and gene expression.

Response to comment 7:

We agree, and we have extensively edited the discussion section to carefully include all the relevant citations mentioned above. In addition, we highlight the focus of this current biophysical study, and we detail the implications of our findings.

Reviewer #2 (Remarks to the Author):

The manuscript by Zheng et al investigates the molecular details of how the Retinoic acid Inducible Gene-I (RIG-I) Receptor interacts with different RNAs and how mutations in RIG-I cause dysregulation. The role of RIG-I and other major intracellular immune receptors in autoimmune disease is important and more information is needed for how they activate and discriminate between viral RNA species and self RNA – and how mutations result in dysregulation and disease. In this work, the authors extend prior work that also made extensive

use of the HDX-MS method, in which they studied the apo state of RIG-I and its binding to ATP, triphosphorylated RNA and longer duplex RNA (Zheng J, Nucleic Acid Research. 2015). That prior work provided a model for how RIG-I (and a related receptor) in the autoinhibited apo-state undergoes allosteric changes when transitioning to an active state. The present manuscript builds on this work and studies the binding of RIG-I (and mutants H830A, C268F and E373A) to different RNAs. The work provides new mechanistic insights into RIG-I regulation. Most interestingly, the results provide a model of how the specific gain-of-function mutations C268F and E373A in RIG-I cause dysregulation leading to Singleton-Merten syndrome, an autoimmune disease.

The experimental work, in particular the HDX-MS component, is very extensive and well-performed. Furthermore, the conclusions of the work are interesting and impactful, also considering that the proposed mechanisms of RIG-I dysregulation by mutations could be relevant for other innate immunity receptors. I am positive towards publication. I have some specific concerns, in particular about the conclusions drawn from the EX1 kinetic data, that need to be addressed prior to publication.

Major comments:

Concerning the observed EX1 kinetics:

Firstly, there are a few sentences detailing EX1 theory that needs to be revised/clarified:

Line 203: The authors write: The CARD2 latch region (Y101-114), which is spatially locked to the HEL2i gate motif in the apo form, displays EX1 exchange behavior upon binding to PAMP RNA, suggesting this region of the receptor is conformationally heterogeneous in solution (4). I think this statement is unclear. The mere presence of EX1 does not show that the receptor conformationally heterogeneous. EX1 informs on the timescale at which structural transitions that facilitate exchange occurs in the protein. A protein undergoing EX2 could be equally conformationally heterogeneous.

Response to comment 1:

We agree. Our interpretation of EX1 data needs more attention, and we need to avoid overinterpreting the observations. This specific sentence has been carefully revised as follows: “The CARD2 latch region (Y101-114), which is spatially locked to the HEL2i gate motif in the apo form, displays EX1 exchange behavior upon binding to PAMP RNA, suggesting that this region undergoes a partial unfolding event and structural transition that facilitates correlated exchange in the protein [5]”.

Line 204: The authors write: “If an unfolding event occurs slow enough for the backbone amide hydrogens within the unfolding region to be fully exchanged, EX1 kinetics is observed with bimodal distribution”. It is when the rate of the refolding or closing event (k_{cl}) is sufficiently slow that EX1 kinetics is observed – please correct the sentence accordingly.
Line 210: The authors write: “Therefore, the region undergoing EX1 kinetics is a mixture of unfolded and folded conformers, which transitions from one state to the other via a slow and correlated exchange event.”. This sentence is misleading, the transition is slow – and thus the exchange is correlated. Please rephrase.

Response to comment 2:

We agree, and we have revised and extended this sentence as follows; “If a refolding event occurs sufficiently slow to allow complete deuterium exchange of backbone amide hydrogens within the unfolding region, then EX1 kinetics are observed [20]. Under EX1 conditions, if an opening or unfolding event involves more than one slow exchanging amide hydrogen, then deuterium exchange occurs simultaneously at these amides. Therefore, a bimodal distribution occurs via a correlated exchange pattern, in which the lower mass envelope corresponds to molecules that have not yet exchanged (not yet unfolded), and the higher mass envelope corresponds to molecules that have undergone exchange (molecules that have unfolded) [5, 20-22]. The region undergoing EX1 kinetics may represent a mixture of unfolded and folded conformers in the same unfolding event. In contrast, EX2 kinetics takes place if the refolding rate is much faster than the intrinsic exchange rate of the amide hydrogens, resulting in one isotopic envelope throughout the labeling time of the experiment.”

Secondly, the authors should be more specific concerning what they can conclude and what

they cannot solely be based on the presence of EX1 kinetics observed for RIG-I. To me the authors lack functional evidence to support that the observed EX1 kinetics (and the folded and unfolded states involved) correspond to the inactive and active forms of RIG-I. I agree that it seems enticing to think so (and very interesting) but that does not necessarily make it so. Do the authors have functional data that show that the derived $t_{1/2}$ values are on the same timescale as the rate of activation or enzymatic activity of RIG-I. If so, such a comparison would make their ultimate conclusions from the EX1 observations much more interesting and convincing.

Response to comment 3:

We agree and apologize for overinterpreting the EX1 data. The observed half-life ($t_{1/2}$) is calculated based on the transition rate from lower MS envelope to higher MS envelope. In a typical experiment HDX behavior is impacted by amide hydrogen bonding, pH, temperature, intrinsic flexibility of the receptor, on-exchange deuterium buffer composition, quench buffer composition, on-exchange time, back exchange post quench, etc. However, all of the differential HDX experiments were performed under identical conditions, pH 7.5, 4 °C, with the same deuterium buffer with 80% deuterium content, and the same chromatography with highly reproducible retention times for peptic peptides. Change in any of these conditions could impact hydrogen bonding of CARD2 latch peptide and thus alter the observed half-life. While different RIG-I receptors (wild-type and mutants) show different half-life values (WT: 13 min, H830A: 17min and E373A: 19 min) upon binding to the same RNA ligand, we can only conclude that this measurement is likely reflective of RIG-I and the mutant receptors switching between inactive and active state. It is unclear how to directly verify this observation in a cell-based assay. For instance, the IFN assay involves incubation of HEK293K cells at 37°C, and the receptor is in a different environment than in the biophysical study. Furthermore, the E373A and C268F mutant RIG-I s are constitutively active upon binding cellular RNAs based on results from the IFN assay, making it difficult to study the transition timescale of RIG-I:RNA complex between active and inactive in cells.

To reduce ambiguities and overinterpretation of results, we selected RNA that had been described in detail in previously published papers for inclusion in our work. All these RNA ligands have been experimentally validated. 3p10I was reported as a minimal RNA duplex that binds to RIG-I in 1:1 ratio that stimulates robust ATPase activity and elicits a RIG-I mediated interferon response in cells [3, 4]. 3p8I was reported as an inactive RNA ligand that still binds to

RIG-I in 1:1 ratio but fails to induce RIG-I mediated interferon response in cells [4, 6]. Cap1-10I was reported as a self RNA bearing m7G cap and 2'-O-methylation, a molecular signature of host mRNA. It binds to RIG-I in 1:1 ratio and the binding affinity drops by 200 folds comparing to 3p10I [3]. Below is a list of the high resolution atomic structures of these RNAs that we used in our study as well as information about the protein-RNA complex. Based on the wealth of published information we selected these RNAs to be part of our comprehensive biophysical study of RIG-I.

3p8I bound Helicase-RD (pdb: 4A2W) [6]

3p10I bound Helicase-RD (pdb: 5F9H) [3]

Cap0-10I bound Helicase-RD (pdb: 5F98) [3]

We selected these RNAs with graded efficacy from a full agonist (3p10I), partial agonist (Cap1-10I), and an inactive RNA (3p8I) that binds the receptor to perturb the conformational dynamics of wild-type and mutant RIG-I receptors. Based on our results, we developed the RIG-I activation model to suggest that the derived half-life values for each protein state (a total of 30 different states) listed in Supplementary Fig. 1a, b, and c correlate and is consistent with the RNA ligand mediated RIG-I agonism described previously. We have revised the relevant sections of the manuscript based on this concern.

For instance:

Line 232: Due to the induced appearance of EX1 kinetics upon RNA binding the authors write “RNA binding by RIG-I drives CARDs module from a closed conformation to a partially opened conformation (Fig. 2a)”.

Response to comment 4:

We have modified this sentence as follows; “This suggests that RNA binding to RIG-I allosterically triggers partial unfolding of the CARDs, and the extent of unfolding in solution is dependent on the efficacy of the specific RNA (**Fig. 2a**).

Line 238: To more precisely measure the increased correlated exchange, we determined the CARDs transition rate from the inactive to the active state”

The presence of EX1 kinetics and the derived $t_{1/2}$ values shows that the rate of unfolding and refolding is slow. But the authors cannot strictly speaking conclude that the slow unfolding/refolding transition observed in the CARDS module is a transition from an inactive to an active state. Additional functional data to support this conclusion is needed, for instance as described above. Alternatively, this caveat needs to be clearly underlined.

Response to comment 5:

We address some of this concern in our response above to comment 3. However, we have carefully revised the sentence: “To more precisely measure the increased correlated exchange, we determined the CARDS transition rate from the lower MS envelope to the higher MS envelope.

Finally on that note, the authors should investigate further if the auto-inhibited apo RIG-I and apo C268F, undergo EX1 kinetics by probing longer timescales (see comment below concerning Fig. 2a). If they do, then one could suppose that these two forms of RIG-I should have some low residual activity, due to the slow build-up of the supposed open and active state, according to the theory put forward by the authors.

Response to comment 6:

As suggested by the reviewer, we have generated data with longer on-exchange time points. Specifically, we now have obtained HDX time points at 3, 5 and 7 hours for apo wild-type RIG-I and apo C268F mutant RIG-I as well as for RIG-I in complex with 3p8I RNA (**Figure A below**). For both apo RIG-I and apo C268F RIG-I, the CARD2 latch peptide shows emerging EX1 kinetics at all of the extended time points (3, 5, and 7 hours). However, the emerging rate of the higher MS envelope in apo RIG-I and apo C268F RIG-I is sufficiently slow that curve fitting analysis fails to calculate the half-life of apo RIG-I and C268F RIG-I in the recorded HDX time points (**see Figure A below**). For apo C268F RIG-I, the overall conformational dynamics are higher than that for apo wild-type RIG-I in solution, as we observe higher deuterium incorporation in apo C268F RIG-I compared to that of apo RIG-I (Supplementary Figure 1a and b, column (i) and (xxiii)).

Figure A. EX1 half-life analysis of longer on-exchange time points. (a) MS spectra of CARD2 latch peptide Y103-114 derived from various protein-ligand complexes at the indicated on-exchange time points (1, 3, 5, 7 hours and Dmax). The abundance of each mass population (high and low) was determined as shown. (b) In each indicated state and time point, the fraction of CARDs molecules in the higher MS population (open conformation) to the total population is plotted against the on-exchange time points (c) Half-life ($t_{1/2}$) of respective partial unfolding event is determined by fitting an exponential curve (as was done in **Figure 2d** of the main manuscript). Predicted RIG-I + 3p8l half-life was 4.97 (Lower 95%=4.69, Upper 95%= 5.26); however, half-life cannot be calculated for either apo RIG-I nor apo C268F RIG-I.

Figure 2a, column 1 + line 221-223. "The EX1 kinetic regime of CARD2 latch peptide in apo RIG-I was difficult to detect as there is very little solvent exchange for the auto-inhibited domain" The Apo state exchanges so little that the presence or absence of EX1 kinetic cannot be determined based on the current data. This can likely be examined by prolonging the time course or maybe increasing the temperature. The current experiments are performed at 4°C. Also, it would appear that an unfolded population occurs at 1 hr for wt RIG-I bound to 3p8l, but not so for apo RIG- thus repeating the exchange experiment at longer times for both state

would be informative and could reveal an effect of 3p8l. Finally, it is not clear to me why the apo-RIG is not simply referred to as wt RIG-I in the figure.

Response to comment 7:

We have also generated new data at longer on-exchange time points for the RIG-I:3p8l RNA complex and calculated the half-life of the CARD2 latch peptide, which resulted in a value of approximately 5 hr (**Figure A panel c** above). We have added this analysis to the revised manuscript and edited the main text to include this new information, plus we include the results in a new **Supplementary Figure 2**. To reveal the effect of 3p8l on RIG-I at longer time points, we performed differential HDX RIG-I with and without 3p8l at 3 and 5 hr exposure to deuterated buffer (**Figure B** below). As shown in Figure B, the HEL1 motif Ic and CTD RNA binding regions show protection to deuterium exchange. Interestingly, CARD2 latch peptide uptakes more deuterium when RIG-I is associated with 3p8l and this de-protection increases at longer HDX time points. We also analyzed the half-life of CARDs turnover in 3p8l bound RIG-I (~ 5 hr), which is much longer compared to that of Cap1 bound RIG-I (~42 min) and 3p10l bound RIG-I (~13 min). Therefore, we conclude that 3p8l has little effect in destabilizing or disrupting the CARD2-HEL2i interface while it still can bind to RIG-I RNA binding motifs in HEL1 and CTD region.

Figure B. HDX perturbation view of RIG-I in the absence and presence of 3p8I RNA at longer on-exchange time points (3 and 5 hr). These new data is presented in Supplementary Fig. 1c in the revised manuscript figures.

Line number 148-155. A reduction on 12 % is mentioned. It is not defined whether it is significant or not. The methods section does not describe how significance is determined - this is defined in the SI, but, in my opinion, this is an important parameter that should be easily accessible in the manuscript and when reviewing the data. Especially, when as all structural discussions are based on the HDX-MS data.

Response to comment 8:

We apologize for not being clear in the method section. We have revised the methods section as follows;

“The intensity weighted mean m/z centroid value of each peptide envelope was calculated and subsequently converted into a percentage of deuterium incorporation. In the absence of a fully deuterated control, corrections for back-exchange were made on the basis of an estimated 70% deuterium recovery, and accounting for the known 80% deuterium content of the deuterium exchange buffer. When comparing the two samples, the perturbation %D is determined by calculating the difference between the two samples. HDX Workbench colors each peptide according to the smooth color gradient HDX perturbation key (D%) shown in each indicated figure. Differences in %D between -5% to 5% are considered non-significant and are colored gray according to the HDX perturbation key [23]. In addition, unpaired t-tests were calculated to detect statistically significant ($p < 0.05$) differences between samples at each time point. At least one time point with a p-value less than 0.05 was present for each peptide in the data set further confirming that the difference was significant.”

HDX perturbation key

Minor comments:

Abstract: Several sentences in the Abstract needs are very long and unclear and must be revised. For instance: “A RIG-I residue (H830) mediates specific sensing of 5' 7-methyl guanosine and 2'O-methylated on the first base (Cap1) self RNA and is coupled with a threefold delay in Caspase Activation and Recruitment Domains (CARDs) partial unfolding event compared to that of 5'ppp RNA”.

Response to comment 9:

The abstract has been revised. Please see the updated abstract.

Line 172 and throughout: the authors use the term “solvent exchange”. To me this is unspecific and strictly speaking incorrect as solvent per se is not exchanged. It could thus confuse a non-expert – I recommend use of a more specific term like HDX, hydrogen exchange or deuterium uptake etc.

Response to comment 10:

The term “solvent exchange” has been changed to read HDX or deuterium exchange.

Figure 1b. Please highlight/bold the R1 and R2 to guide the reader.

Response to comment 11:

We have modified and highlighted R1 and R2 in red accordingly.

Figure 1c. Suggestion – add the origin of the different RNA constructs. e.g. Self-RNA or Viral RNA to the figure. This will make it easier for readers not familiar with the research area to follow the argumentation.

Response to comment 12:

We have added one sentence in the Figure 1b legend as follows: 3p10I and Cap1-10I are representative of viral and self-RNAs, respectively.

Line number 504. Missing letter in fraction, should be fraction.

Response to comment 13:

Corrected.

Line number 148-150. "The HDX data obtained for sequence overlapping peptides were consolidated to individual amino acid values using a residue averaging approach (29)". For people without former knowledge of HDX, it could sound like they have residue-resolved HDX-data everywhere. This is probably not the case. Please elaborate. Also, for all overlapping peptides for which this procedure was applied, the authors should inspect the maximal-labeled control and verify that the a similar back-exchange was observed. The Schriemer lab has observed that overlapping peptides can exhibit large differences in back-exchange rendering subtractive analysis problematic for those peptides.

Response to comment 14:

We agree and thank the reviewer for this comment. We have revised the manuscript as follows;

"Digestion optimization for HDX studies resulted in greater than 90% sequence coverage for full-length WT RIG-I (~100kDa protein) as well as the gain-of-function RIG-I mutants H830A, E373A and C268F (**Supplementary Fig. 1 a, b and c**). In the absence of fragmentation data from electron capture dissociation (ETD), residue level deuteration data were approximated by using HDX data from overlapping peptides, and consolidating these data using a residue averaging approach previously described [24] (**Supplementary Fig. 1b**). These data were calculated and mapped onto the PyMol structure model using HDX Workbench [23]."

This single residue consolidation with smooth coloring approach was applied in our recent study wherein we looked at differential HDX experiments of 26 conformational states of VDRRXR heterodimer upon binding to different compounds, DNAs and co-activator proteins to aid in visualization of large datasets [25].

Additional comments: In our HDX experiment, we used 5 μ l of protein sample and diluted it into 20 μ l of deuterium containing on-exchange buffer such that the on-exchange sample (5 μ l sample + 20 μ l D₂O) contains 80% deuterium content by volume. We have examined one maximally labelled control sample (H830A RIG-I) and used HDX Workbench to calculate the deuterium recovery (%) of all the analyzed peptides (see Figure C below). The average %D in this Dmax sample was approximately 70% with a range of 50%-80% (Figure C). While our platform is not fully maximized to reduce deuterium loss post quench, the sample is maintained at low pH and all of the solvents, syringes, valves, columns (except the protease column which is at 15°C) are maintained at 4°C within a cold box. The transfer tube from the deli frig to the mass spectrometer is as short as possible and insulated, and the ion source conditions are carefully set to balance deuterium loss with ion signal. With this set up we routinely obtain 70% Average, 50%-80% range deuterium recovery. Importantly, we have shown in a recent publication the reproducibility of this platform [26]. Regardless, given that all of the experiments are differentials run within the same day, the deuterium loss is expected to be equivalent for peptides under condition A versus those from condition B. Although 70% deuterium recovery is an estimate and the percentage may vary from peptide to peptide, we can use a single correction factor within HDX Workbench or use the values from the Dmax experiment [23]. This is a generally well-accepted approach for differential HDX experiments in which both samples are treated identically (with the same LC gradient, pH 2.4, 0 °C).

Figure C. Maximum deuterium recovery calculation for RIG-I peptides under the operating conditions used in this study.

In addition, we generated Dmax data on the H830A mutant receptor. Using this new information, we directly compared the HDX data set (H830A RIG-I +/- 3p10l) that was calculated using the single 70% deuterium recovery value that that corrected using the Dmax control (Figure D below). The results obtained from the two different data processing approaches are comparable as shown below in **Figure D**.

Figure 2a, column 2 + line 223-226: "In contrast, CARDs (1-228) protein, where the CARDs domain is not auto-inhibited and fully exposed to solvent, resulted in rapid deuterium incorporation and the latch peptide underwent EX2 exchange indicating structural homogeneity in solution".

To me, this does not look like pure EX2 kinetics. The peak width at 10min and 15min is markedly larger than at 30min and 1h. Hinting at a possible mixture of kinetics (EXX). Furthermore, the apo CARD state is defined as "open", but its conformation is not similar to the "open" state described with blue in columns 4-9, as it takes the apo CARDs around 30min to be fully deuterated.

Response to comment 15:

We thank the reviewer for this comment and we agree. We have corrected the figure heading and replaced EX2 with 'absence of EX1'. In the text, the sentence has been revised as follows; 'In contrast, analysis of the CARDs (1-228) protein, where the CARDs domain cannot be auto-inhibited, resulted in significant deuterium incorporation and the same latch peptide showed absence of EX1 kinetics in the recorded HDX time points'.

Also, HDX analysis reveals that the isolated apo CARDs domain (CARDs only) is prone to significantly higher deuterium incorporation when compared to CARDs that is sequestered by HEL2i such as in full-length apo receptors (WT, H830A, E373A and C268F) (supplementary Figure 1a and b, column (ii), (i), (xii), (xv) and (xxiii)). We thus conclude that apo CARDs adopts open conformation in solution.

Line 439-440. "HDX MS data was calculated with the in-house developed software and corrected for back-exchange on an estimated 70% recovery."

Figure 2a shows that the authors have recorded 'maximally labelled' samples. Did they normalize the deuterium incorporation to these peptide specific values or to the average back-exchange reported to be 30%. I would strongly recommend the former. Please elaborate on the procedure used and why.

Response to comment 16:

Please see response to comment 14 above.

Figure 3.d-g Non-linear or logarithmic x-axis on graph, d-g. Please remove the s in the single time point labels on the x-axis.

Response to comment 17:

We have corrected this error.

Supporting Information: The HDX-MS experimental section has some cases of incorrect nomenclature, spellings and unexplained abbreviations etc. For instance, gHCL, D20 etc. Please revise carefully and consistently.

Response to comment 18:

We have corrected this error.

References cited in this response letter.

1. Lassig, C., et al., *Correction: ATP hydrolysis by the viral RNA sensor RIG-I prevents unintentional recognition of self-RNA*. Elife, 2016. **5**.
2. Malathi, K., et al., *Small self-RNA generated by RNase L amplifies antiviral innate immunity*. Nature, 2007. **448**(7155): p. 816-9.
3. Devarkar, S.C., et al., *Structural basis for m7G recognition and 2'-O-methyl discrimination in capped RNAs by the innate immune receptor RIG-I*. Proc Natl Acad Sci U S A, 2016. **113**(3): p. 596-601.
4. Kohlway, A., et al., *Defining the functional determinants for RNA surveillance by RIG-I*. EMBO Rep, 2013. **14**(9): p. 772-9.
5. Zheng, J., et al., *High-resolution HDX-MS reveals distinct mechanisms of RNA recognition and activation by RIG-I and MDA5*. Nucleic Acids Res, 2015. **43**(2): p. 1216-30.
6. Luo, D., et al., *Visualizing the determinants of viral RNA recognition by innate immune sensor RIG-I*. Structure, 2012. **20**(11): p. 1983-8.
7. Schuberth-Wagner, C., et al., *A Conserved Histidine in the RNA Sensor RIG-I Controls Immune Tolerance to N1-2'O-Methylated Self RNA*. Immunity, 2015. **43**(1): p. 41-51.
8. Lassig, C., et al., *ATP hydrolysis by the viral RNA sensor RIG-I prevents unintentional recognition of self-RNA*. Elife, 2015. **4**.

9. Kato, H. and T. Fujita, *RIG-I-like receptors and autoimmune diseases*. *Curr Opin Immunol*, 2015. **37**: p. 40-5.
10. Lu, C. and M. MacDougall, *RIG-I-Like Receptor Signaling in Singleton-Merten Syndrome*. *Front Genet*, 2017. **8**: p. 118.
11. Lassig, C. and K.P. Hopfner, *Discrimination of cytosolic self and non-self RNA by RIG-I-like receptors*. *J Biol Chem*, 2017. **292**(22): p. 9000-9009.
12. Zhang, H.X., et al., *Rig-I regulates NF-kappaB activity through binding to Nf-kappab1 3'-UTR mRNA*. *Proc Natl Acad Sci U S A*, 2013. **110**(16): p. 6459-64.
13. Jang, M.A., et al., *Mutations in DDX58, which encodes RIG-I, cause atypical Singleton-Merten syndrome*. *Am J Hum Genet*, 2015. **96**(2): p. 266-74.
14. Yoneyama, M., et al., *The RNA helicase RIG-I has an essential function in double-stranded RNA-induced innate antiviral responses*. *Nat Immunol*, 2004. **5**(7): p. 730-7.
15. Wu, B. and S. Hur, *How RIG-I like receptors activate MAVS*. *Curr Opin Virol*, 2015. **12**: p. 91-8.
16. Peisley, A., et al., *Structural basis for ubiquitin-mediated antiviral signal activation by RIG-I*. *Nature*, 2014. **509**(7498): p. 110-4.
17. Hou, F., et al., *MAVS forms functional prion-like aggregates to activate and propagate antiviral innate immune response*. *Cell*, 2011. **146**(3): p. 448-61.
18. Peisley, A., et al., *RIG-I forms signaling-competent filaments in an ATP-dependent, ubiquitin-independent manner*. *Mol Cell*, 2013. **51**(5): p. 573-83.
19. Wu, B., et al., *Molecular Imprinting as a Signal-Activation Mechanism of the Viral RNA Sensor RIG-I*. *Molecular Cell*, 2014. **55**(4): p. 511-523.
20. Fang, J., J.R. Engen, and P.J. Beuning, *Escherichia coli processivity clamp beta from DNA polymerase III is dynamic in solution*. *Biochemistry*, 2011. **50**(26): p. 5958-68.
21. Yang, B., et al., *Vps4 disassembles an ESCRT-III filament by global unfolding and processive translocation*. *Nat Struct Mol Biol*, 2015. **22**(6): p. 492-8.
22. Trelle, M.B., et al., *Local transient unfolding of native state PAI-1 associated with serpin metastability*. *Angew Chem Int Ed Engl*, 2014. **53**(37): p. 9751-4.
23. Pascal, B.D., et al., *HDX workbench: software for the analysis of H/D exchange MS data*. *J Am Soc Mass Spectrom*, 2012. **23**(9): p. 1512-21.
24. Keppel, T.R. and D.D. Weis, *Mapping residual structure in intrinsically disordered proteins at residue resolution using millisecond hydrogen/deuterium exchange and residue averaging*. *J Am Soc Mass Spectrom*, 2015. **26**(4): p. 547-54.
25. Zheng, J., et al., *HDX reveals the conformational dynamics of DNA sequence specific VDR co-activator interactions*. *Nat Commun*, 2017. **8**(1): p. 923.
26. Cummins, D.J., et al., *Two-Site Evaluation of the Repeatability and Precision of an Automated Dual-Column Hydrogen/Deuterium Exchange Mass Spectrometry Platform*. *Anal Chem*, 2016. **88**(12): p. 6607-14.

REVIEWERS' COMMENTS:

Reviewer #2 (Remarks to the Author):

The authors have in their revision provided extensive new HDX-MS data and details concerning their data analysis. This new information adequately address my concerns.

One comment - concerning the new sentence:

"The region undergoing EX1 kinetics may represent a mixture of unfolded and folded conformers in the same unfolding event."

The sentence is still not clear and stating that a mixture of unfolded and folded conformers exist is not really informative...that would, strictly speaking, always be the case for any protein in solution. It is rather the relative proportions of unfolded vs folded states and the rates at which they interconvert that can be revealed by a more detailed analysis of EX1 kinetics.

Reviewer #3 (Remarks to the Author):

In this study Zheng, Wang et al. describe the effect of a pair of disease-associated mutations (C268F and E373A) in the dsRNA innate immune sensor RIG-I on the conformational flexibility, signaling activity, and ATPase activity of RIG-I. A mutation that allows RIG-I to bind capped cellular RNAs, H830A, is also examined. Conformational flexibility and solvent exposure of the CARD signaling domains, which correlate tightly with signaling activation, are quantified by hydrogen-deuterium exchange (HDX). ATPase, signaling and RNA binding activities are were assayed in vitro using biochemical and cell-based assays. The authors report that the H830A mutation

Although similar studies with WT RIG-I have been published previously (ref. 4), the present study provides some useful insights on how the C268F and E373A cause disease by constitutively signaling in the absence of infection. Moreover, the HDX data highlight differences in how each mutant dysregulates RIG-I signaling. Both mutations decouple signaling from ATP hydrolysis but C268F appears to do so by affecting the conformational flexibility of RIG-I whereas E373A affects the ATP-dependent proofreading activity of RIG-I. Both mechanisms are distinct from that of H830A, which affects RNA binding specificity.

Most of the concerns of the first two reviewers seem to have been adequately addressed (see below for an exception). The principal remaining concerns are regarding the interpretation that the CARDS "partially unfold" and that the manuscript is poorly organized, does not flow logically in places and is therefore difficult to follow overall.

Major concerns:

1. The text, and specifically the organization of the Results section are not very logically organized, making the story difficult to follow. The text frequently skips from one mutant to another and from one activity to another, specific arguments are for the most part not organized into separate paragraphs, and subheadings in the Results section do not accurately or informatively reflect the content below them. There are several redundant passages, some passages in the Results belong in the Discussion, and the Discussion could be developed further. Specific examples are listed below under "Minor comments". Cumulatively, this adds up to a major concern, however, as the mechanistic models for how C268F and E373A work, and how they differ from each other and what the outstanding questions are, remain somewhat unclear.
2. The authors state repeatedly that the CARDS unfold or partially unfold upon dsRNA binding, but HDX does provide a direct or accurate measure of protein folding, and it seems that unfolding probably only really occurs in the short CARD2 latch peptide region (residues 101-114). It has

been well established that the CARDS must associate with MAVS CARD to form as stably folded oligomeric assembly in order to activate RIG-I-dependent signaling. Hence the references to partial unfolding are potentially misleading. This also applies to Fig. 5 where partial unfolding is listed as an intermediate between the closed and open conformations. "Partially open" would be preferable, but the authors should probably simply delete the term throughout.

3. The only reviewer comment that is not satisfactorily addressed is Comment 4 of Reviewer 2 regarding MAVS CARD binding. The ability of the RIG-I variants to bind MAVS CARD is highly pertinent to the mechanism of signaling activation and the purification of MAVS CARD has been described previously (Ref. 19) and appears relatively straightforward.

Minor points.

1. Abstract Lines 38-40. The meaning of this sentence is unclear and ambiguous. Please rewrite and remove or qualify the reference to partial unfolding.
2. Results, Lines 142 and 166. These subtitles appear completely disconnected from the text that follows them. Please replace with more accurate and descriptive subtitles, which together will provide a logical flow to the text. The subtitle at Line 166 should be moved to line 217 as a new subtitle, with the CARD2 latch peptide referenced explicitly instead of "CARDS partial unfolding events". A new subtitle should be written for the section lines 166-216, referencing the CTD and cap recognition.
3. Line 292: it is unclear what the authors mean by "CARDS turnover" in this subtitle. A more specific and descriptive subtitle should be written
4. Lines 312-381 (or similar) should be split into a different subheading.
5. Lines 294-355- this section is particularly difficult to follow. Clarify and state more succinctly what the differences are between WT and the 373 mutant.
6. Lines 326-330 belong in the Discussion (consolidate with related sentence already present in Discussion).
7. Lines 338-342- this important point should be further emphasized, perhaps by starting a new paragraph with Line 343.
8. Lines 387-417- should be split up into more than one paragraph, with different aspects discussed in different paragraphs.
9. Lines 415-422 belong in the Discussion.
10. Discussion: the first paragraph should be moved to the introduction, and any redundant points should be deleted. Please discuss how it the C268F mutation might cause the CARDS to be released by both self- and non-self RNAs, in the context of previously published structural data. Please comments on why is might be that E373A drives release of CARDS with both capped and uncapped RNAs while at the same time retaining the ability to "distinguish" between the two types of ligand- this point comes across as contradictory.
11. Lines 44, 362, 463: deleted "much".
12. Fig. 1a. Please show the CARD2 latch region.
13. Fig. 5 is difficult to interpret. Can it be simplified to ease interpretation? What is increasing along the x-axis of each panel? The x-axis labels should be used as titles for the diagrams and a new common label should be added, such as "Signaling activity". "Partial unfolding" should be removed from the y axis. It would be helpful if the molecular diagrams in Panel b could be modified to highlight how the 373 and 268 mutants differ in their signaling mechanisms, and also to reflect how partially open (partially unfolded) species differ from open or closed species. Also it would seem to make more sense if the yellow stars for ATP were present somewhere in panel in the two instances where ATP hydrolysis is shown to occur (only ADP is shown).

Response to Reviewers:

Reviewer #2 (Remarks to the Author):

The authors have in their revision provided extensive new HDX-MS data and details concerning their data analysis. This new information adequately addresses my concerns.

One comment - concerning the new sentence:

"The region undergoing EX1 kinetics may represent a mixture of unfolded and folded conformers in the same unfolding event."

The sentence is still not clear and stating that a mixture of unfolded and folded conformers exist is not really informative...that would, strictly speaking, always be the case for any protein in solution. It is rather the relative proportions of unfolded vs folded states and the rates at which they interconvert that can be revealed by a more detailed analysis of EX1 kinetics.

Author's Response:

We thank the reviewer for their time and effort to make our manuscript stronger. The sentence mentioned above has been changed as follows: the region undergoing EX1 kinetics could reveal the rate at which the relative proportions of unfolded and folded conformers interconvert.

Reviewer #3 (Remarks to the Author):

In this study Zheng, Wang et al. describe the effect of a pair of disease-associated mutations (C268F and E373A) in the dsRNA innate immune sensor RIG-I on the conformational flexibility, signaling activity, and ATPase activity of RIG-I. A mutation that allows RIG-I to bind capped cellular RNAs, H830A, is also examined. Conformational flexibility and solvent exposure of the CARD signaling domains, which correlate tightly with signaling activation, are quantified by hydrogen-deuterium exchange (HDX). ATPase, signaling and RNA binding activities were assayed in vitro using biochemical and cell-based assays. The authors report that the H830A mutation

Although similar studies with WT RIG-I have been published previously (ref. 4), the present study provides some useful insights on how the C268F and E373A cause disease by constitutively signaling in the absence of infection. Moreover, the HDX data highlight differences in how each mutant dysregulates RIG-I signaling. Both mutations decouple signaling from ATP hydrolysis but C268F appears to do so by affecting the conformational flexibility of RIG-I whereas E373A affects the ATP-dependent proofreading activity of RIG-I. Both mechanisms are distinct from that of H830A, which affects RNA binding specificity.

Most of the concerns of the first two reviewers seem to have been adequately addressed (see below for an exception). The principal remaining concerns are

regarding the interpretation that the CARDS “partially unfold” and that the manuscript is poorly organized, does not flow logically in places and is therefore difficult to follow overall.

Major concerns:

1. The text, and specifically the organization of the Results section are not very logically organized, making the story difficult to follow. The text frequently skips from one mutant to another and from one activity to another, specific arguments are for the most part not organized into separate paragraphs, and subheadings in the Results section do not accurately or informatively reflect the content below them. There are several redundant passages, some passages in the Results belong in the Discussion, and the Discussion could be developed further. Specific examples are listed below under “Minor comments”. Cumulatively, this adds up to a major concern, however, as the mechanistic models for how C268F and E373A work, and how they differ from each other and what the outstanding questions are, remain somewhat unclear.

Author's Response:

We thank the reviewer for this comment and apologize for the improper organization of subheadings, and redundancies in the Results and Discussion sections. We have modified the respective sections specifically listed under minor comments to improve the readability of this manuscript. The discussions for mechanistic model for C268F and E373A mutations has been modified and updated. We also added a discussion regarding to the structural mechanisms that drive the signaling events by RIG-I variants.

2. The authors state repeatedly that the CARDS unfold or partially unfold upon dsRNA binding, but HDX does provide a direct or accurate measure of protein folding, and it seems that unfolding probably only really occurs in the short CARD2 latch peptide region (residues 101-114). It has been well established that the CARDS must associate with MAVS CARD to form as stably folded oligomeric assembly in order to activate RIG-I-dependent signaling. Hence the references to partial unfolding are potentially misleading. This also applies to Fig. 5 where partial unfolding is listed as an intermediate between the closed and open conformations. “Partially open” would be preferable, but the authors should probably simply delete the term throughout.

Author's Response:

We have replaced the term “partially unfolding” to “partially opening” throughout the text and modified Figure 5 accordingly to avoid any confusion.

3. The only reviewer comment that is not satisfactorily addressed is Comment 4 of Reviewer 2 regarding MAVS CARD binding. The ability of the RIG-I variants to bind MAVS CARD is highly pertinent to the mechanism of signaling activation and the purification of MAVS CARD has been described previously (Ref. 19) and appears relatively straightforward.

Author's Response:

We thank the reviewer for this comment. As we previously stated the RIG-I CARDS MAVS CARD interaction is complicated, involving RIG-I CARDS tetramer formation (assisted by Lysin63 linked poly-ubiquitin chains) and nucleation of RIG-I CARDS and MAVS CARD complex assemble. We believe this is very important and should be part of a study focused on downstream signaling activation following CARDS opening. However, this study is outside the scope of the current manuscript. We do look forward to studying the MDA5 CARDS assemble with and without MAVS CARD, and compare with that of RIG-I CARDS-MAVS CARD complex.

Minor points.

1. Abstract Lines 38-40. The meaning of this sentence is unclear and ambiguous. Please rewrite and remove or qualify the reference to partial unfolding.

Author's Response:

The term partial unfolding is a term used to describe the observation of EX1 exchange behavior. However, for clarity we have changed it to "partial opening."

2. Results, Lines 142 and 166. These subtitles appear completely disconnected from the text that follows them. Please replace with more accurate and descriptive subtitles, which together will provide a logical flow to the text. The subtitle at Line 166 should be moved to line 217 as a new subtitle, with the CARD2 latch peptide referenced explicitly instead of "CARDS partial unfolding events". A new subtitle should be written for the section lines 166-216, referencing the CTD and cap recognition.

Author's Response:

We thank the reviewer for such comment. The subheadings have been modified as "RNA chemical features and IFN- β activities" and added new subheadings as followings: "Cap1 discrimination by CTD-HEL RNA recognition module" and "RNA surveillance is coupled with CARDS partial opening".

3. Line 292: it is unclear what the authors mean by "CARDS turnover" in this subtitle. A more specific and descriptive subtitle should be written

Author's Response:

We thank the reviewer for such comment. We have changed the subheading as "E373A affects the ATP-dependent proofreading of RIG-I". What is more, CARDS turnover has been replaced with CARDS transition.

4. Lines 312-381 (or similar) should be split into a different subheading.

Author's Response:

We added a new subheading as "E373A affects the ATP-dependent proofreading of RIG-I".

5. Lines 294-355- this section is particularly difficult to follow. Clarify and state more succinctly what the differences are between WT and the 373 mutant.

Author's Response:

We apologize for the poor readability of this paragraph. We have reorganized the sentences in a more logical way to highlight the major differences between WT and E373A variant RIG-I.

6. Lines 326-330 belong in the Discussion (consolidate with related sentence already present in Discussion).

Author's Response:

We have noted such redundancy and consolidated the sentence in the discussion section.

7. Lines 338-342- this important point should be further emphasized, perhaps by starting a new paragraph with Line 343.

Author's Response:

We have started a new line in the modified text.

8. Lines 387-417- should be split up into more than one paragraph, with different aspects discussed in different paragraphs.

Author's Response:

We have split the text into three paragraphs.

9. Lines 415-422 belong in the Discussion.

Author's Response:

We have noted such redundancy and consolidated the sentence in the discussion.

10. Discussion: the first paragraph should be moved to the introduction, and any redundant points should be deleted. Please discuss how it the C268F mutation might cause the CARDS to be released by both self- and non-self RNAs, in the context of previously published structural data. Please comments on why is might be that E373A drives release of CARDS with both capped and uncapped RNAs while at the same time retaining the ability to “distinguish” between the two types of ligand- this point comes across as contradictory.

Author's Response:

We have made the necessary modifications.

11. Lines 44, 362, 463: deleted “much”.

Author's Response:

Deleted.

12. Fig. 1a. Please show the CARD2 latch region.

Author's Response:

We have modified the figure.

13. Fig. 5 is difficult to interpret. Can it be simplified to ease interpretation? What is

increasing along the x-axis of each panel? The x-axis labels should be used as titles for the diagrams and a new common label should be added, such as “Signaling activity”. “Partial unfolding” should be removed from the y axis. It would be helpful if the molecular diagrams in Panel b could be modified to highlight how the 373 and 268 mutants differ in their signaling mechanisms, and also to reflect how partially open (partially unfolded) species differ from open or closed species. Also it would seem to make more sense if the yellow stars for ATP were present somewhere in panel in the two instances where ATP hydrolysis is shown to occur (only ADP is shown).

Author’s Response:

We thank the reviewer for suggestions to improve Figure 5. We have modified the figure. We have removed “partial unfolding.” We labeled the x-axis as “functional RIG-I checkpoints and signaling activities” in panel a and “dysregulated RIG-I checkpoints and signaling activities” in panel b. With respect to how partially open species differ from open or closed species, we have shown the quantitative analysis of CARDS EX1 exchange kinetics (T half-life) to illustrate the different extent of opening rate associated with each signaling RIG-I conformer. Finally, we have highlighted the ATP molecule in the updated figure.